# SPARSE UPCYCLING: TRAINING MIXTURE-OF-EXPERTS FROM DENSE CHECKPOINTS

**Aran Komatsuzaki**[*†]
Georgia Institute of Technology

**Joan Puigcerver**[†]
Google Research

**James Lee-Thorp**[†]
Google Research

**Carlos Riquelme**
Google Research

**Basil Mustafa**
Google Research

**Joshua Ainslie**
Google Research

**Yi Tay**
Google Research

**Mostafa Dehghani**
Google Research

**Neil Houlsby**
Google Research

## ABSTRACT

Training large, deep neural networks to convergence can be prohibitively expensive. As a result, often only a small selection of popular, dense models are reused across different contexts and tasks. Increasingly, sparsely activated models, which seek to decouple model size from computation costs, are becoming an attractive alternative to dense models. Although more efficient in terms of quality and computation cost, sparse models remain data-hungry and costly to train from scratch in the large scale regime. In this work, we propose sparse upcycling – a simple way to reuse sunk training costs by initializing a sparsely activated Mixture-of-Experts model from a dense checkpoint. We show that sparsely upcycled T5 Base, Large, and XL language models and Vision Transformer Base and Large models, respectively, significantly outperform their dense counterparts on SuperGLUE and ImageNet, using only $\sim 50\%$ of the initial dense pretraining sunk cost. The upcycled models also outperform sparse models trained from scratch on $100\%$ of the initial dense pretraining computation budget.[1]

## 1 INTRODUCTION

Increased scale is one of the main drivers of better performance in deep learning. From BERT (Devlin et al., 2019) to GPT-3 (Brown et al., 2020) to PaLM (Chowdhery et al., 2022) in natural language processing, or from AlexNet (Krizhevsky et al., 2017) to ViT-G (Zhai et al., 2022) in vision, breakthroughs in performance have been obtained from larger hardware, datasets, and architectures. This trend holds true in many other domains too, including speech (Baevski et al., 2020), reinforcement learning (Schrittwieser et al., 2020), multimodal learning (Yu et al., 2022), and scientific applications of deep learning (Jumper et al., 2021).

However, most state-of-the-art neural networks are trained from-scratch; that is, starting from randomly initialized weights. The cost for training such networks is growing rapidly. For example, in language, BERT-Large (345M parameters, proposed in 2018) required an estimated $0.5$ ZFLOPS to train, while GPT-3 (175B parameters, from 2020) required $314$ ZFLOPS (Brown et al., 2020), and PaLM (540B parameters, from 2022) required $2527$ ZFLOPS (Chowdhery et al., 2022). As a result of these computation costs, research into new large language models is often limited to a small number of teams with access to lots of resources. To enable significant further progress, we must develop cheaper ways of training giant models.

In this paper, we explore *model upcycling*: upgrading an existing model with a relatively small additional computational budget. In particular, we focus on upcycling dense models into larger, sparsely activated Mixture-of-Experts (MoEs). We do not use any new unique sources of data (Wei

---

[*]Work done while interning at Google Research.

[†]Contacts: `aran1234321@gmail.com`, {`jpuigcerver`, `jamesleethorp`}`@google.com`.

[1]Code is available at `https://github.com/google-research/vmoe` (Vision) and `https://github.com/google-research/t5x/tree/main/t5x/contrib/moe` (Language).

et al., 2021; Ouyang et al., 2022). We assume the existence of a pretrained dense Transformer checkpoint (e.g. (Wolf et al., 2020)), that we then use to warm-start the training of a MoE. By leveraging the additional capacity of from the MoE layers, we obtain an MoE model more performant than the original model, at a smaller cost than was used to train the original model. Across all model sizes that we study for both language and vision, with less than $40\%$ additional budget, upcycling improves the network's performance beyond what would be achieved by continued training the original Transformer model.

Sparse upcycling may be particularly valuable in two scenarios: (i) One has access to a pretrained Transformer (there are many publicly available) and wants to improve it with a modest or constrained computational budget. (ii) One is planning to train a large model, and do not know whether a dense or MoE model would be more effective (the latter often being more performant, but more technically challenging to train): one can have both by first training the dense model, then upcycling it into an MoE model once the dense model saturates.

A central challenge in model upcycling is overcoming the initial performance decrease entailed by changing a trained network's structure. We present a model surgery recipe that is effective in both vision and language, and numerous ablations for the key components that make it work well. In experiments on Vision Transformers (Dosovitskiy et al., 2021) and T5 language models (Raffel et al., 2020), we show that upcycling is highly effective when the computation budget lies between +10% and +60% of the cost to train the original (dense) network. For example, increasing the performance of ViT-B/16 by at least 1% on ImageNet 10-shot requires an additional 58% extra training time (relative to the original checkpoint) if we continue training the dense model; however, it only takes 13% extra training time with the upcycled version. Similarly, upcycled T5-Large and T5-Base models outperform their dense counterparts by 1.5-2 absolute points on SuperGLUE using 46% and 55% extra training, respectively.

## 2 BACKGROUND

In this section we recap of the main components used in sparse upcycling: Transformer-based language and vision models, and sparsely activated Mixture-of-Experts (MoEs).

### 2.1 SPARSELY ACTIVATED MIXTURE-OF-EXPERTS (MOE)

Dense models apply all parameters to every input. Accordingly, growing the model capacity results in increased computational cost. Sparse models attempt to alleviate this fundamental issue by only activating a subset of parameters for each input. Sparsely activated Mixture-of-Experts (MoE) models are an accelerator friendly family of sparse models that allow training of models with up to trillions of parameters (Shazeer et al., 2017; Fedus et al., 2022).

MoE models typically alternate standard dense Transformer blocks with MoE blocks. In particular, we usually replace the MLPs in a Transformer block with a number of "experts" (typically themselves MLPs) with different learnable parameters and a router—a small neural network—that decides which expert is applied to each individual token. A number of routing algorithms have been developed, for example Top-K (Shazeer et al., 2017), BASE and Sinkhorn-BASE layers (Lewis et al., 2021; Clark et al., 2022), Hash layers (Roller et al., 2021), and Expert Choice routing (Zhou et al., 2022).

We generally focus on Expert Choice routing, which works as follows. Let $E$ denote the total number of experts in a MoE layer, and $n$ the total number of tokens. The router outputs a matrix $\mathbf{R} \in \mathbb{R}^{n \times E}$ with the routing probabilities, where row $r_i \in \mathbb{R}^E$ corresponds to the $i$-th token and is a distribution over $E$ experts ($r_{ij} \geq 0$ and $\sum_j r_{ij} = 1$). Then, every expert $e$ independently chooses the $T$ tokens with highest probabilities for $e$ (i.e., we perform top-T per column) and processes them. We parameterize $T$ as $T = C(n/E)$, where $C$ is a capacity factor that we control to choose more or fewer tokens per expert. When $C = 1$, each expert processes exactly $n/E$ tokens; note that some tokens may be processed by several experts, while others by none. This allows for a model parameter count increase with minimal FLOPs overhead.[2] Letting $C > 1$ usually leads to higher performance at a higher compute cost.

---

[2]The FLOPs overhead comes from the (relatively modest) router computation of $\mathbf{R}$.

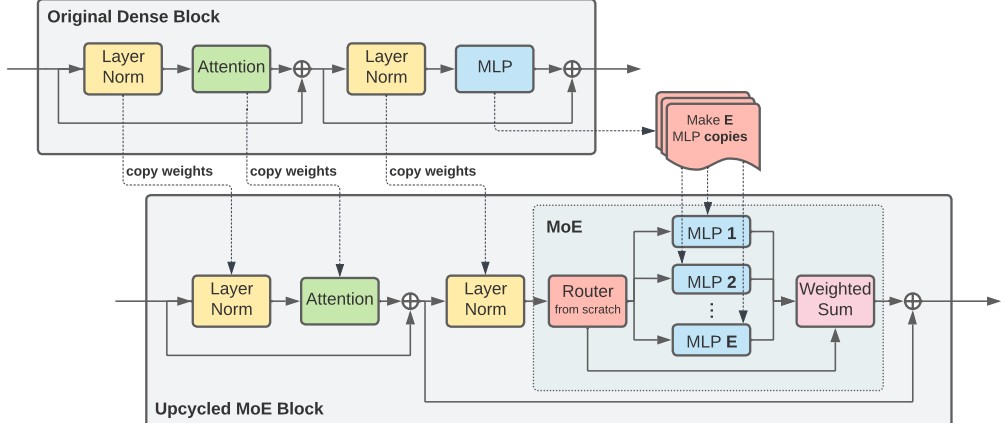

Figure 1: The upcycling initialization process. All parameters, and optionally their optimizer state, are copied from the original checkpoint, except those corresponding to the MoE router, which does not exist in the original architecture. In particular, the experts in the new MoE layer are identical copies of the original MLP layer that is replaced.

## 2.2 ARCHITECTURES

We apply the same sparse upcycling recipe to both language and vision tasks, focusing on the T5 (encoder-decoder) (Raffel et al., 2020; Narang et al., 2021) and Vision Transformer (encoder) (Dosovitskiy et al., 2021) architectures, respectively. We generally adopt the same gating function and MoE hyperparameters in the encoders of both models. See Section 3.1 for specific design choices and Appendix A for differences between the vision and language upcycling setups.

**Vision**. Vision Transformers (ViT) are encoder-only Transformer architectures (Liu et al., 2021; Radford et al., 2021; Touvron et al., 2021; He et al., 2022) which tokenize and embed images. We upcycle models based on the B/32, B/16, L/32 and L/16 variants. The resultant MoEs broadly follow Vision MoE Transfomers ("V-MoE") (Riquelme et al., 2021), with two differences; we perform global average pooling (Zhai et al., 2022) and use Expert Choice routing.

**Language**. We experiment with the T5 (Raffel et al., 2020) encoder-decoder as our archetypal language model. We upcycle the Base, Large, and XL variants of the model. We sparsify both the encoder and decoder. As in our vision setup, the model encoder applies Expert Choice routing. We use Top-K routing in the decoder with $K = 2$; see also Section 3.1.

## 3 THE UPCYCLING ALGORITHM

The algorithm is illustrated in Figure 1. To upcycle a model, we need a dense model's parameters (i.e. a *checkpoint*). The number and shape of Transformer blocks in the new model is identical to that in the original dense model. A subset of the of the MLP layers are expanded into MoE layers. The remaining MLP layers, along with all of the layer-norm and attention layers, and the embedding and output layers are copied across from the original model to the new model. Each MoE layer contains a fixed number of experts. Each expert is initialized as a copy of the original MLP. In addition, we add a router whose weights are randomly initialized. In Section 4.2.2, we experiment with different variations on this basic recipe. After the new model is loaded and initialized, we continue training it for a number of additional steps depending on the available budget and resources. We use the original hyperparameters: same batch size, learning rate schedule, and weight decay leading to the original checkpoint; see also Appendix A for full training details.

### 3.1 DESIGN DECISIONS

An upcycled model's performance is heavily influenced by the configuration of the MoE layers. Increasing the model capacity by increasing the number of upcycled layers, number of experts or expert capacity will generally lead to a higher quality model, but will also increase the computational cost and/or result in a greater initial quality drop, due to the more drastic reconfiguration of the layers.

**Router type**. For upcycled vision models and for the encoder of upcycled language models, we use Expert Choice routing with capacity factor $C = 2$. To avoid train time (full batch teacher forcing) versus inference time (single token auto-regressive decoding) discrepancies, we use Top-K ($K = 2$) routing in the language decoder. In Section 4.2.2, we show that Expert Choice routing outperforms standard Top-K routing for upcycling, while both beat dense continuations.

**Number layers to upcycle.** Adding more MoE layers increases the model capacity dramatically, at the expense of increasing the model's cost, and also causing the quality of the upcycled model to initially drop further relative to the original dense model. Based on our ablation in Section 4.2.2 and prevailing conventions in the MoE literature (Lepikhin et al., 2021), unless otherwise specified, we replace half of the MLP layers in our upcycled models with MoE layers.

**Number of experts to add in upcycled layers**. Each new expert provides new learnable parameters that extend the model capacity. The expert capacity—the number of tokens expert processes–is inversely proportional to the number of experts, thus adding more experts does not significantly affect the FLOPS or the run time of the model. However, with a very large number of experts, the upcycled model experiences a larger initial quality drop relative to the baseline dense model. Given sufficient upcycling compute, this initial drop can be overcome. In our studies, we upcycle with +20% to +100% of the initial dense baseline model's computational cost, and in this regime we find that 32 experts provides a good compromise. We explore varying the number of experts in Section 4.2.2.

**Expert capacity**. By tuning the expert capacity, $C$, we control the number of experts that process each token on average.[3] Larger expert capacity generally yields larger quality but also increases the FLOPS and run time. Although increasing the expert capacity yields quality gains on a per step basis, we find that $C = 2$ generally offers good quality on a compute time basis. We ablate through different capacity factors in Section 4.2.2.

**Resuming optimizer state (vision only)**. When upcycling a model, we can resume the optimizer state from the original dense checkpoint together with the model parameters. In Appendix B.6, we find that reusing the optimizer state gives a performance boost for vision models. We did not, however, see any improvement from reusing the dense model optimizer state in our language experiments, so we only reuse the optimizer state for vision models.

**Normalize weights after routing (vision only)**. In an effort to reduce the performance drop when applying the upcycling model surgery, we attempted to normalize the router combine weights of each token to $1$. This follows that intuition that each token was previously only processed by a single "expert" MLP in the dense model. Appendix B.7 shows that router weight normalization helps upcycled vision models, but hurts the performance of upcycled language models. One hypothesis for this different behavior is that the vision models use Expert Choice routing everywhere, but the language models use Expert Choice in the encoder and Top-K routing in the decoder.

## 4 EXPERIMENTS

In this section, we present the main experimental results of the paper. We also share the takeaways of a number of ablations aimed at identifying the key aspects of our algorithm; full results are included in Appendix B. Most of the results are presented as quality vs. cost plots, where we use the upstream or downstream performance to measure quality, and training time in terms of TPU-core-days (as prominent cost metrics (Dehghani et al., 2021)) or training steps (when the cost per step is the same for all the compared models) to measure computation cost.

---

[3]For Expert Choice routing, more capacity means that each expert can choose more tokens. For standard Top-K routing, more capacity means that each token is more likely to fit into the buffer of its desired expert.

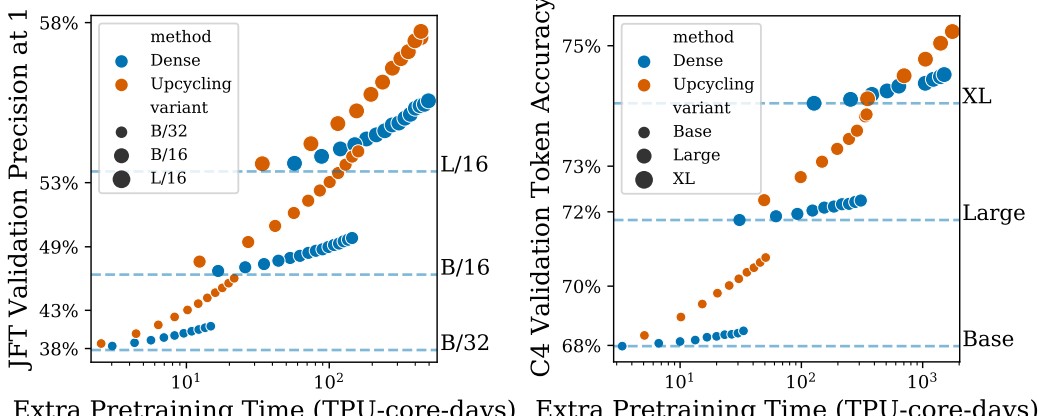

Figure 2: Pretraining performance achieved by the dense continuation and upcycling methods, for different Transformer variants. The left plot shows the performance on the vision task and the right one on the text task. The x-axis shows the extra pretraining time (TPU-core-days), with respect to the total time needed to train the original dense checkpoints, for each size. The horizontal lines indicate the quality (y-axis) of the original dense checkpoints.

## 4.1 EXPERIMENTAL SETUP

All upcycled experiments begin from a pretrained dense model checkpoint. Because all of our starting dense checkpoints are trained with an inverse square root learning rate schedule, training can be continued without discontinuities in the learning rate schedule. We upcycle models and continue training, showing performance for varying amounts of continued training. As a baseline, we also continue the training of the original dense model ("dense continuation").

**Vision experiments**. MoE Vision Transfomers ("V-MoE") models are trained broadly following the protocol of Riquelme et al. (2021). Upstream pretraining is done on JFT300M (Sun et al., 2017), with validation metrics computed on a held-out set of 894,574 examples. Few-shot transfer follows Dosovitskiy et al. (2021), whereby a least-squares regressor predicts one-hot classes given frozen image representations. We further validate our results on ImageNet using 10-shot – i.e. 10 training examples per class. We do this for 5 different training sets, and report average accuracy across them. For full finetuning, we replace the pretraining head with a randomly initialized head, and finetune the entire network. See Appendix A.2.2 for further details.

**Language experiments**. Our language experiments follow the setup of Raffel et al. (2020): we pretrain using the span corruption task on the English C4 dataset (Raffel et al., 2020) and finetune on a proportional mix of all SuperGLUE (Wang et al., 2019) tasks simultaneously. We include specific training details in Appendix A.2, but highlight one important aspect here: For Base model sizes, for which we perform the majority of our ablations, we pretrain the dense baseline starting checkpoint ourselves. To highlight the versatility of our upcycling algorithm, for Large and XL models, we instead begin all experiments from official T5 1.1 checkpoints (Narang et al., 2021; Roberts et al., 2022).

## 4.2 EXPERIMENTAL RESULTS

### 4.2.1 CORE RESULTS

Figure 2 shows a detailed comparison of upstream metrics of upcycled models and dense continuation models at various model sizes both for vision (left panel) and language (right panel). For any given model size and task, we observe that the dense and upcycled models perform close to each other when we apply a very limited extra training budget – indeed, close to their discontinuous horizontal line representing the original checkpoint's performance. Once we apply a non-trivial amount of extra compute, a clear pattern emerges showing the strong gains delivered by the upcycled architecture.

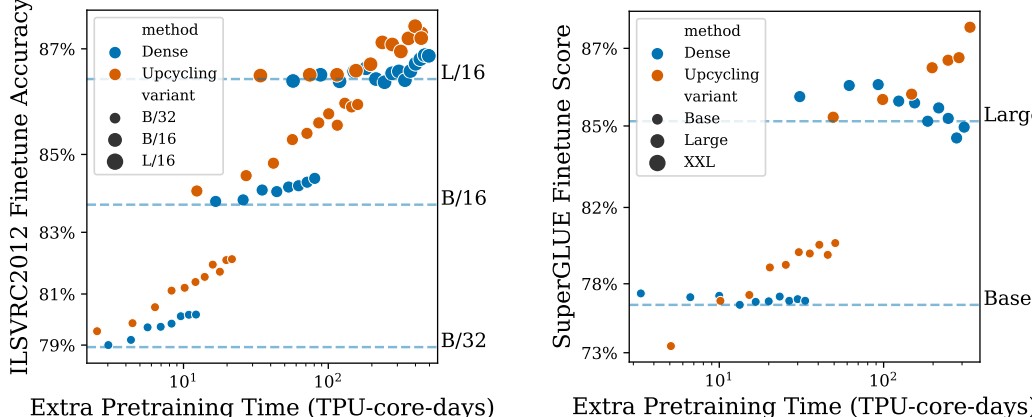

Figure 3: Full finetuning performance achieved by the dense continuation and upcycling methods. The left plot shows the performance on ImageNet and the right one on SuperGLUE tasks. The x-axis shows the extra pretraining time (TPU-core-days), with respect to the total time needed to train the original dense checkpoints, for each size. The horizontal lines indicate the quality (y-axis) of the original dense checkpoints.

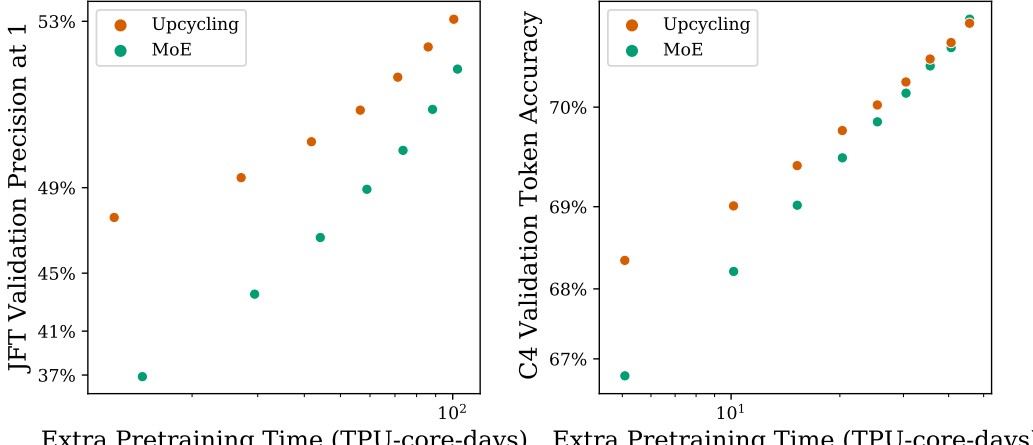

Figure 4: Pretraining performance achieved by the upcycling method and a MoE model trained from scratch, for B/16 (left plot, vision task) and Base (right plot, text task). The x-axis shows the extra pretraining time (TPU-core-days), with respect to the total time needed to train the original dense checkpoints. The MoE model trained from scratch only catches up to the language upcycled model after about 120% of the original dense checkpoint computation budget (second last orange and green dots from the right).

Figure 3 shows the performance after finetuning the models trained in Figure 2. For vision (left panel), the upstream performance gains generally transfer fairly cleanly downstream. For language (right panel), there is more variance in the performance after finetuning. Nevertheless, the trend clearly favors the upcycled language models.

Figure 4 compares sparse upcycling with sparse models trained from scratch. As training from scratch does not reuse the computation cost already sunk into the dense checkpoint, it takes longer, on a extra train time basis, to catch up with the upcycled models. The language MoE model trained from scratch requires about 120% of the original dense checkpoint's computation budget to catch up to the upcycled model. The relatively faster quality gains, on a per step basis, of the MoE models trained from scratch can be attributed to the relatively larger learning rate and that the experts are able to independently develop and diversify from the beginning. Figure 4 suggests that, given a very large computation budget ($> 100\%$ of the initial dense model's computation budget), the MoE-from-

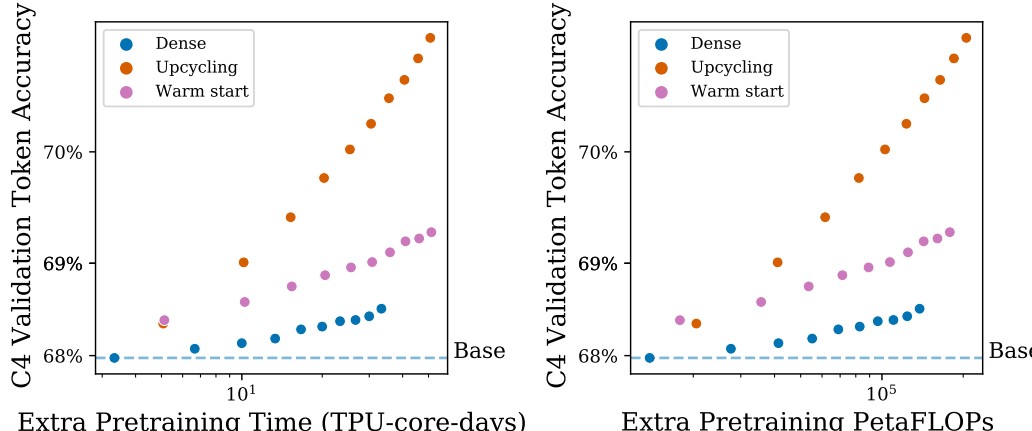

Figure 5: Pretraining performance achieved by sparse upcycling and dense upcycling from a T5 Base checkpoint (text task). The x-axes show the extra pretraining time (TPU-core-days) and extra pretraining (Peta)FLOPs, with respect to the the original dense checkpoints. Following recommendations in (Rae et al., 2021), we only increase the number of layers when warm starting ("depth tiling") to roughly match the runtime of the sparse model.

scratch model will eventually catch the upcycled model. For such large computation regimes, it may be preferable to train MoE models from scratch. For constrained or limited compute budgets ($< 100\%$ of the initial computation budget), sparse upcycling is a more efficient use of resources.

Finally, Figure 5 compares sparse upcycling with warm starting ("dense upcycling"). We warm start larger models from the dense Base checkpoint by replicating new layers ("depth tiling") in the same tiling patterns as in (Rae et al., 2021). The densely upcycled models quickly see gains over the original dense checkpoint, but underperform the sparse model. We did not attempt to increase the model hidden dimensions ("width tiling"), which (Rae et al., 2021) found to be less effective.

### 4.2.2 ABLATIONS

In this section we summarize important architecture and training ablations relative to the baseline model described in Section 3. Full results are provided in Appendix B. Unless stated otherwise, vision ablations use a B/16 sparse model with 32 experts, $C = 1$ and 6 MoE layers placed in the last few blocks of the model. The dense checkpoint was trained for 14 epochs, and we train for an additional 7 epochs (up to a total of 21 epochs). For our language ablations, our default configuration is unchanged: we use a Base model with 32 experts, $C = 2$ and 6 MoE layers interspersed throughout the model. We train for between 0.5 million and 1 million extra steps.

**Amount of dense pretraining**. The upcycling efficiency may, in principle, depend on how converged the initial dense model is. To explore this, in Figure 6, we upcycle a B/16 vision model starting from different dense checkpoints with varying amounts of pretraining. From a given dense checkpoint, we compare upcycling and dense continuation for 200k steps. Independent of when we start upcycling, the performance improvement from doing so is fairly consistent.

**Router type**. While our default upcycling recipe uses Expert Choice routing (in the encoder), the same recipe can be applied to other routing mechanisms. For vision, Appendix B.1 shows that although Top-K routing, with Batch Prioritized Routing (BPR) (Riquelme et al., 2021), matches Expert Choice routing performance on a per step basis, as it is slightly slower, Top-K routing underperforms Expert Choice routing on a per train time basis. Note both approaches beat dense.

**Expert capacity factor**. The more tokens processed per expert, the greater the amount of compute per input example and (generally) the higher the model quality. Our sparse models smoothly control this via the expert capacity factor $C$. Appendix B.2 explores how the performance-speed trade-off varies as a function of $C$. Although increasing the expert capacity yields quality gains on a per step basis, we find that $C = 2$ generally offers the best quality on a compute time basis, for both language and vision models.

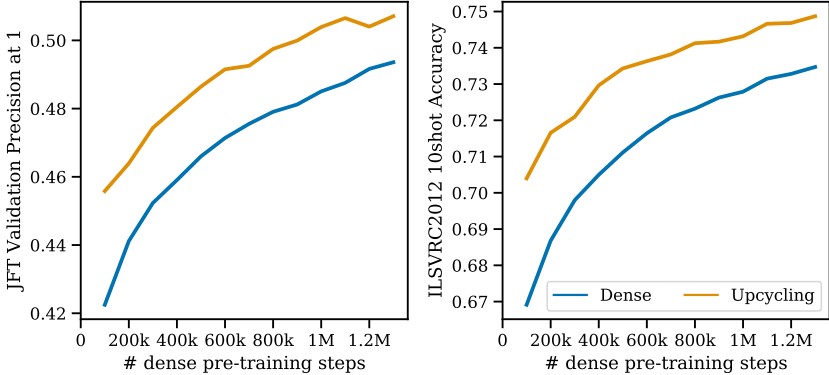

Figure 6: Upcycling performance as a function of the amount of pretraining steps for the original dense checkpoint. The y-axis shows the performance after 200k steps of further training on top of the original dense checkpoint, for both the dense continuation and upcycled models. The x-axis shows for how long the original dense checkpoint was trained. The gains from upcycling are fairly consistent independent of the amount of initial pretraining. Note: for this particular ablation, we use a capacity factor of $C = 1$, to ensure that the FLOPS and run times of the dense model and sparsely upcycled model are roughly comparable on a per step basis.

**Number of MoE layers**. A key decision when upcycling a model is how many sparse layers to add. Model capacity and parameter-count increases with the number of MoE layers, but at the expense of slowing down model run time. Appendix B.4 provides an ablation over the number of MoE layers in an upcycled B/16 vision model. In this case, around 6 MoE layers (out of a total of 12 layers) offers the best precision quality, although fewer sparse layers lower the compute costs.

**Initialization of experts**. The standard upcycling recipe copies and replicates the dense MLP to each expert. As the router directs different tokens to each expert, the experts will start to diverge from one another, and from their initial MLP weights. Appendix B.5 compares the standard recipe with randomly initializing the experts (i.e. train them from scratch). For limited computation budgets, randomly initializing experts underperforms the standard recipe; for larger compute budgets it eventually matches the standard upcycling recipe performance.

We also tried copying the original MLP weights and adding independent (Gaussian) noise to each expert, in an effort to promote more expert diversity. Adding too much noise when copying MLP weights into experts hurts performance, while adding small amounts of noise has little to no effect; see also Appendix B.9.

**Number of experts**. Adding more experts increases the number of model parameters and, up to a point, the quality of the model. Given that the number of tokens each expert processes is inversely proportional to the number of experts (see Section 2.1), adding more experts does not significantly affect the model FLOPS nor its running time. However, for a very large number of experts, the upcycled model may experience a larger initial quality drop relative to the baseline dense model. Appendix B.3 explores this trade-off and shows that, at least for Base sized models, more experts usually yield better performance.

## 5    RELATED WORK

**Reuse of trained parameters**. Prior work has focused on speeding up training through a warm start by reusing parameters of an existing model. Berner et al. (2019) explore ideas of reusing the previous edition of a trained model during a long training process using an under-development environment. Given a trained model, Net2Net (Chen et al., 2015) propose a function-preserving initialization to warm start training a deeper or wider model. Recently, Gopher (Rae et al., 2021) also explored warm starting larger models from smaller models in a large compute training regime and show that the larger, warm started model can converge to a quality comparable to that of the equivalent model trained from scratch. In Section 4.2.1, we show that warm starting significantly underperforms sparse upcycling. Yang et al. (2021); Lin et al. (2021) show that they can reduce the number of training

iterations with models that inititally share parameters across layers (Lan et al., 2019; Dehghani et al., 2018) but gradually unshare (or "delink") the parameters while training.

In an effort to reduce total training cost, several works explore progressively growing models during training (Gong et al., 2019; Dong et al., 2020; Li et al., 2020; Shen et al., 2022). The core idea is to decompose the training process into stages, each of which apply growth operators to increase the model size from the previous stage by copying weights or stacking new layers on top. In some cases, each training stage will only update parameters of the new layers, which saves the cost of a full backward computation (Yang et al., 2020). Gu et al. (2020) show that compound scaling (scaling depth, width and input length together) is favorable and propose a strategy with various growing operators on each dimension.

Sparse upcycling, which we introduce in this paper, follows a similar motivation. However, unlike the above works, we focus on compute regimes that are a fraction of the original model's training. We also present a recipe for growing a trained dense model to a sparse model, instead of a larger dense model. This enables us to enjoy the extra capacity due to increased parameters, while maintaining the inference cost due to the sparsity of computation.

**Pruning**. Pruning is typically employed as a post-training architecture search to construct smaller and faster models from larger models (LeCun et al., 1989; Gale et al., 2019; Blalock et al., 2020). However, "dynamic pruning" (Evci et al., 2020) has also been used during training to find sparser architectures from dense models. Similar to pruning, sparse upcycling also introduces sparsity to a dense model, however, unlike pruning, we grow the existing dense models into a larger sparse model.

**Sparsely-activated Mixture-of-Experts (MoE)**. In this work, we sparsify existing dense models into MoE models. MoE models (Shazeer et al., 2017) offer the promise of increasing model scale (parameter count) with sublinear increases in computation cost (FLOPS). Recently, there has been a growing number of MoE works achieving state-of-the-art quality and remarkable efficiency gains on both language and vision tasks (Lepikhin et al., 2021; Fedus et al., 2022; Riquelme et al., 2021; Artetxe et al., 2021; Du et al., 2021; Zoph et al., 2022; Mustafa et al., 2022). All of these models are large and trained from scratch with randomly initialized weights and fixed architectures.

Several MoE works have also attempted to improve upon typical training algorithms by adapting or "evolving" the model architecture during training. Nie et al. (2021) progressively sparsify MoE layers during training by slowly adjusting the gating function from a "dense setup", where all tokens are routed to all experts, to a fully "sparse setup", where tokens are only routed to a subset of experts. Zhang et al. (2022) observed that, for most inputs, only a small fraction of Transformer MLP activations are nonzero. Based on this, they propose sparsification procedure that splits the parameters of MLP blocks into multiple experts and add a routing mechanism. Similarly, Zuo et al. (2022) split up the MLPs in a pre-trained dense model into multiple experts to form a sparse model for fine-tuning. Closely related to our work, Wu et al. (2022) present a novel algorithm to sparsify dense models in the context of finetuning on detection and segmentation tasks. Similar to Nie et al. (2021), the initial performance drop when training on the original dataset is avoided by applying a *dense* mixture of experts in the forward pass. However, at our target large scales, simultaneously activating all experts for each token is not feasible. Finally, Gururangan et al. (2022) adapt sparse, domain expert language models to new domains by initializing a new domain expert from the most probable existing expert under the domain posterior distribution.

## 6 CONCLUSIONS

Training large neural networks on huge datasets has proven to be a remarkably successful trend in deep learning research, especially in recent years. It has also proven to be very computationally expensive. Pretrained models are now widely available, thus making it possible for many practitioners to further finetune and adapt fixed model architectures on their data of interest. However, significant progress requires providing more flexibility in adapting and improving the model architecture itself.

We proposed a simple recipe to reuse pretrained dense checkpoints to initialize more powerful sparse models. Our algorithm leverages the pretrained model compute and weights, and provides a smooth transition to sparsely activated Mixture-of-Experts models that offer more capacity and flexibility at inference. We presented experimental results both for vision and language models at various scales; these evidence large performance gains relative to continuing to the dense model. Our ablations

highlight the importance of careful algorithmic choices, and suggest key aspects to consider when trying to find good performance-cost trade-offs for specific compute budgets.

Transfer learning and prompt tuning is becoming increasingly popular, and for good reason. It allows the reuse and tuning of models by a larger body of researchers and practitioners that may only have access to limited computational and data resources. Accordingly, we believe that techniques aimed at growing existing models, compartmentalizing or freezing submodules, replicating and then decoupling model components, and finally smoothly resuming training after model surgery, will prove essential for a dynamic ecosystem of models. We summarize such process as *upcycling*, and offer a first instance in the context of sparse models. We look forward to new extensions and improvements on this simple idea.

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

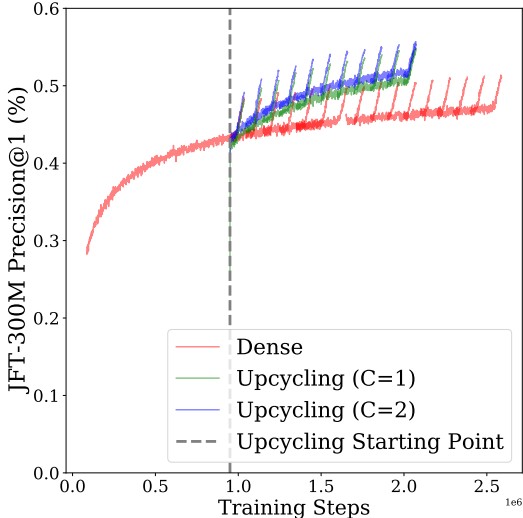

Figure 7: Effect of upcycling a VIT-B/16 model after 14 epochs of dense training. We show a number of cooldowns (decreasing learning rate to zero) for each model, in case that is the maximum training budget available. The difference in slope for the dense and upcycling training curves is significant.

# A  TRAINING AND EVALUATION DETAILS

In this section, we describe the precise setup for our language and vision experiments. Figure 7 illustrates the type of combined training curves we obtained before and after upcycling.

## A.1  UPSTREAM TRAINING

### A.1.1  LANGUAGE

We first pretrain a dense Base model from scratch for 1 million steps with a batch size of 512 sequences. We use the Adafactor optimizer with an inverse square root decay and a peak learning rate of $0.01$. This results in plateauing performance for the dense Base model. Upcycled models are then initialized from the 1 million step dense checkpoint, and compared, on a compute time basis, with further training of the dense model ("dense continuation"); see Figure 2 in the main text. To highlight the versatility of our upcycling algorithm, for Large and XL models, we instead begin all experiments from official T5 1.1 checkpoints (Narang et al., 2021).[4]

We use the same hyperparameters for the upcycled model as for the corresponding dense model that we initialized from, continuing the inverse square root learning rate schedule where the dense checkpoint left off. For all sizes, every *other* layer was upcycled, using 32 experts, starting with the second layer. Similar to dense pretraining, we do not include any dropout (or expert dropout; see Section A.2.1 below). Router parameters are initialized randomly with a zero-mean normal distribution with standard deviation $0.02$. We use a maximum routing group size of 4096 tokens. For Top-2 routing (in the decoder) we include an auxiliary MoE loss, with scaling factor $0.01$, to ensure tokens are distributed more evenly across all experts in the decoder (Shazeer et al., 2017; Fedus et al., 2022).

Upcycling was performed on TPU v4 accelerators using 64 chips for Base and Large and 256 chips for XL. All sizes used expert partitioning, but only XL used model partitioning, with 4 partitions.

---

[4]For experiments starting from the official checkpoints we match the official, larger batch size (2048), to ensure no discontinuities in continuing the dense baseline pretraining.

### A.1.2 VISION

Dense ViT models are pretrained on JFT300M Sun et al. (2017). We train with Adafactor (Shazeer & Stern, 2018), and decoupled weight decay (magnitude 3 on head and 0.03 on body) following Zhai et al. (2022). We use a batch size of 4096. The learning rate schedule consists of a learning warmup of 10 000 steps, followed by reverse square root decay with timescale 100 000 steps and ending with a linear cooldown to 0 over 50 000 steps. We use a fixed peak learning rate of $4 \cdot 10^{-4}$. [5] Models employing a patch size of 32 (i.e. B/32, L/32) were trained for a total of 14 epochs, while those employing a patch size of 16 (i.e. B/16, L/16) were trained for 28 epochs. By default, we begin upcycling half-way through the total number of epochs in each case. This roughly corresponds to the number of epochs used to train the corresponding variant in ViT (Dosovitskiy et al., 2021). For B/16, for example, we assume that a dense checkpoint trained for 14 epochs is given, and we either continue training or apply upcycling for another 14 epochs.

## A.2 MODEL TRANSFER

### A.2.1 LANGUAGE

For finetuning on SuperGLUE, we generally adopt the conventional setup (Raffel et al., 2020; Narang et al., 2021) where we finetune on all SuperGLUE tasks, in a proportional mix, for 200K steps with a batch size of 128. Each example has input length 512 and target decoding length of 62. In our figures, we report the average SuperGLUE accuracy score across 3 runs for each data point.

For finetuning Dense models on SuperGLUE, we use Adafactor with the default, constant learning rate of $10^{-3}$ and a dropout rate of 0.1 (Raffel et al., 2020; Narang et al., 2021). For finetuning upcycled models, because there are many more parameters, it can be helpful to increase the dropout rate for the experts (Fedus et al., 2022), while using the default dropout rate of 0.1 for all "dense" parameters. For upcycled Base models, we obtained the strongest results for a constant learning rate of $10^{-4}$ with an expert dropout rate of 0.1. For upcycled Large models, we found slightly stronger results with a learning rate of $10^{-3}$ and an expert dropout rate of 0.3. Decreasing (or increasing) the learning rate was not helpful for the Dense Base or Large models.

### A.2.2 VISION

**Few-shot linear evaluation**. The fewshot evaluation poses classification as a linear regression task, where inputs are frozen representations computed by a pretrained model, and outputs are one-hot vectors representing the ground truth (Dosovitskiy et al., 2021). There are two key changes compared to prior works which used this method (Dosovitskiy et al., 2021; Riquelme et al., 2021):

- Multiple seeds. The evaluation involves randomly selecting $N$ examples per class. To reduce dependency on that choice, we run 5 random seeds, and report the average test accuracy across them.
- Fixed L2 regularization. Prior works considered a range of L2 regularizations. The optimal value was picked based on average test accuracy across all datasets considered for few-shot evaluation. We fix the L2 regularisation at 1024.

**Full finetuning**. We finetune models on ImageNet2012 using SGD and a batch size of 512. We use a cosine decay learning rate schedule with a linear warmup. We sweep over two training schedules: (i) 5k steps, with a warmup of 200 steps, and (ii) 10k steps, with a warmup of 400 steps. Alongside this we sweep over learning rates [0.1, 0.03, 0.01, 0.003, 0.001, 0.0003]. For each pretrained model, there are therefore 12 finetuning sweeps; we select based on optimal validation accuracy, and report the corresponding test accuracy.

## A.3 MODEL PARAMETERS

Table 1 gives the number of parameters for models used in the main text.

---

[5] Note that this is slightly different to ViT (Dosovitskiy et al., 2021), which changing the learning rate slightly based on the model variant.

Table 1: Model sizes. The number of parameters for sparsely upcycled and MoE-from-scratch models are the same (both are of type "Sparse"). The number of parameters is also unchanged between different routing mechanisms.

| Modality | Model | Type | Fraction of MoE Layers | # Experts | # Parameters |
|---|---|---|---|---|---|
| Vision | B/32 | Dense | – | – | 101M |
| Vision | B/16 | Dense | – | – | 100M |
| Vision | L/32 | Dense | – | – | 324M |
| Vision | L/16 | Dense | – | – | 322M |
| Vision | B/32 | Sparse | 6 / 12 | 32 | 980M |
| Vision | B/16 | Sparse | 6 / 12 | 32 | 978M |
| Vision | L/32 | Sparse | 12 / 24 | 32 | 3.44B |
| Vision | L/16 | Sparse | 12 / 24 | 32 | 3.44B |
| Language | Base | Dense | – | – | 248M |
| Language | Large | Dense | – | – | 783M |
| Language | XL | Dense | – | – | 2.85B |
| Language | Base | Sparse | 6 / 12 | 32 | 2.00B |
| Language | Large | Sparse | 12 / 24 | 32 | 7.22B |
| Language | XL | Sparse | 12 / 24 | 32 | 26.26B |

## A.4 PARALLELIZATION STRATEGIES

Sparsely activated Mixture-of-Experts (MoE) models combine three types of parallelization strategies to train large models across multiple accelerator chips: data, model and expert parallelism. We use data parallelism to shard the training batch across devices. We use expert parallelism to partition experts across devices; for example, placing experts 1 and 2 on device 1, experts 3 and 4 on device 2, and so on. Model parallelism is a third axis along which model weights (matrices) can be sharded across devices; for example, expert 1 is split across devices 1 and 2, expert 2 is split across devices 3 and 4, and so on. Model parallelism is beneficial for scaling to larger model sizes. See also (Fedus et al., 2022) for a more detailed discussion of these three parallelization strategies.

## B ABLATIONS AND ADDITIONAL EXPERIMENTS

In this section, we present results for a number of model ablations that try to identify good choices for the main upcycling algorithm decisions. As mentioned in the main text, unless stated otherwise, vision ablations use a B/16 sparse model with 32 experts, $C = 1$ and 6 MoE layers placed in the last few block of the model. The dense checkpoint was trained for 14 epochs, and we train for an additional 7 epochs (up to a total of 21 epochs). Note that, for $C = 1$, comparing performance on a per step basis is a reasonably close approximation of a comparison on a per train time basis.

For our language ablations, our default configuration is unchanged: we use a Base model with 32 experts, $C = 2$ and 6 MoE layers interspersed throughout the model. We train for between 0.5 million and 1 million extra steps.

### B.1 ROUTER TYPE

While our default upcycling recipe uses Expert Choice routing Zhou et al. (2022) (in the encoder), the same recipe can be applied to other routing mechanisms. Here, we compare with Top-K routing Shazeer et al. (2017), which is a very popular alternative. Table 2 shows that, for vision, sparse upcycling with Top-K routing works comparably well to Expert Choice, on a per step basis, provided we also use Batch Priority Routing (BPR) (Riquelme et al., 2021). BPR sorts tokens according to a model confidence proxy so that –when experts are full– high confidence tokens are given priority. We suspect this may be helpful right at the beginning, when applying the upcycling, to avoid discarding important tokens. Expert Choice avoids this problem by design, as experts are always balanced and select the most 'relevant' tokens.

Table 2: Sparse Upcycling on L/32 vision models with Expert Choice and Top-K routing (also known as Top-K). $K$ refers to the number of selected experts per token, while $C$ refers to the capacity factor. Notice that with Expert Choice routing, each token chooses $C$ experts on average. The initial dense checkpoint was trained for 7 epochs. Note that these comparison are on a per-step basis, and that Expert Choice upcycled models are actually slightly faster than Top-K models; see Figure 8.

| Model | Capacity | From | Extra Epochs | Val Prec@1 | ImageNet 10shot |
|---|---|---|---|---|---|
| Dense | – | Dense | 7 | 49.60 | 73.59 |
| Expert Choice | $C = 1$ | Dense | 7 | 51.91 | 74.04 |
| Top-K | $K = 1$ | Dense | 7 | 51.51 | 74.40 |
| Expert Choice | $C = 2$ | Dense | 7 | 52.80 | 74.83 |
| Top-K | $K = 2$ | Dense | 7 | 52.88 | 74.91 |
| Expert Choice | $C = 1$ | Scratch | 7 | 50.42 | 72.95 |
| Expert Choice | $C = 2$ | Scratch | 7 | 51.28 | 74.01 |
| Expert Choice | $C = 1$ | Scratch | 14 | 54.84 | 75.02 |
| Expert Choice | $C = 2$ | Scratch | 14 | 55.46 | 75.75 |

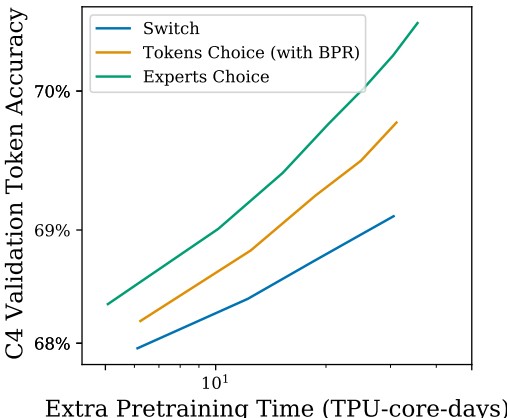

Figure 8: Comparison of Expert Choice, Top-2 and Switch (Top-1) routing mechanisms for a Base upcycled language model.

For language, similar ablations (Figure 8) shows that Expert Choice routing outperforms both Top-2 routing (with BPR) and switch (Top-1) routing, on a train time basis.

## B.2 ROUTER CAPACITY FACTOR

Sparsifying the dense model increases the model capacity (number of parameters). However, if the capacity factor $C = 1$, then the FLOPS is very similar to the original, dense model (modulo the small routing costs). We can increase the per-token compute by increasing $C$. Figure 9 investigates this, and shows our results for vision (left and center panels) and language (right panel).

For vision, we see that extreme values ($C = 1$ and $C = 5$) underperform intermediate values ($C = 2$ and $C = 3$) that offer better trade-offs. For language, the trend is even stronger: A capacity factor of $C = 2$ stands out as the best option on a per compute basis.

## B.3 NUMBER OF EXPERTS

Adding more experts increases the number of model parameters and, up to a point, the quality of the model. Given that the number of tokens each expert processes is inversely proportional to the number of experts (see Section 2.1), adding more experts usually only leads to very modest computational (and wall time) overheads. However, for a very large number of experts, the upcycled model may experience a larger initial quality drop relative to the baseline dense model.

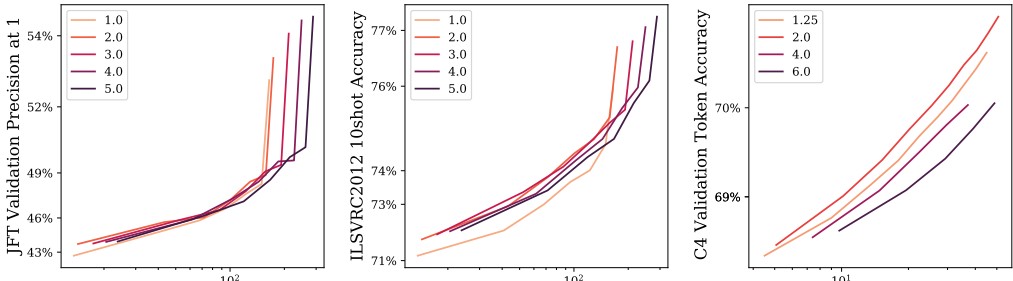

Figure 9: Pretraining performance achieved by upcycling using different capacity factors, for a B/16 (left and center panels, vision tasks) and a Base T5 model (right plot, text task). The x-axis shows the extra pretraining time (TPU-core-days), with respect to the total time needed to train the original dense checkpoint. Although using a bigger capacity factors can result in an absolute better performance when runtime is disregarded (e.g. see the vision results), for a given fixed compute budget, it is usually better to use a capacity factor of around 2.0.

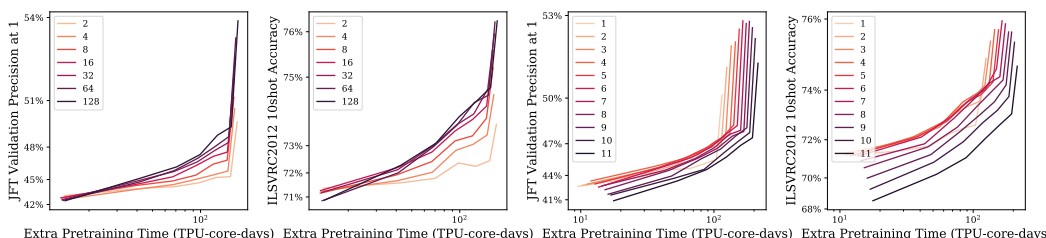

Figure 10: Pretraining performance achieved by the upcycling method on the vision tasks, using different number of experts per MoE layer (two left plots, with a total number of 6 MoE layers), and a different number of MoE layers (two right plots, with a total number of 32 experts; all MoE layers are placed at the top Transformer blocks). The x-axis shows the extra pretraining time (TPU-core-days), with respect to the total time needed to train the original dense checkpoint.

Figure 10 (two left panels) shows the results of a vision experiment with 6 MoE layers with a number of experts ranging from 2 to 128. For a fixed amount of compute (value in the x-axis), we see that more experts is generally better for this B/16 model. Figure 11 shows the final metric values both for upstream (JFT precision at 1) and downstream (ImageNet 10-shot) with respect to the number of experts. We see steady improvements upstream, and –at some point– diminishing returns downstream.

### B.4 NUMBER OF MoE LAYERS

Another key decision is how many layers to sparsify. More layers leads to higher model capacity, while –especially for higher $C$– it introduces significant extra wall time overhead. We ablate this for vision models, as shown in Figure 10 (two right panels). For a B/16 model with 12 blocks, we train upcycled versions with an increasing number of MoE layers; MoE layers are consecutive and start from the last layer. For example, the model labeled as '5' corresponds to a model where the last 5 MLP layers are sparsified, and so on. Thus, model '1' only has one MoE layer (the last one) and it is the computationally cheapest in terms of wall time. We do not include a model where all layers are sparsified (would correspond to '12') as we found that sparsifying the very first block tends to be problematic.

We see in Figure 10 (two right panels) that more MoE layers is not always better even on a per step basis; see Figure 12 for both upstream and downstream metrics. Looking at a fixed value of the $x$-axis in Figure 10 (right panels), we conclude that something between Last-5 and Last-6 (40-50% of layers sparsified) offers the most attractive trade-off in this case.

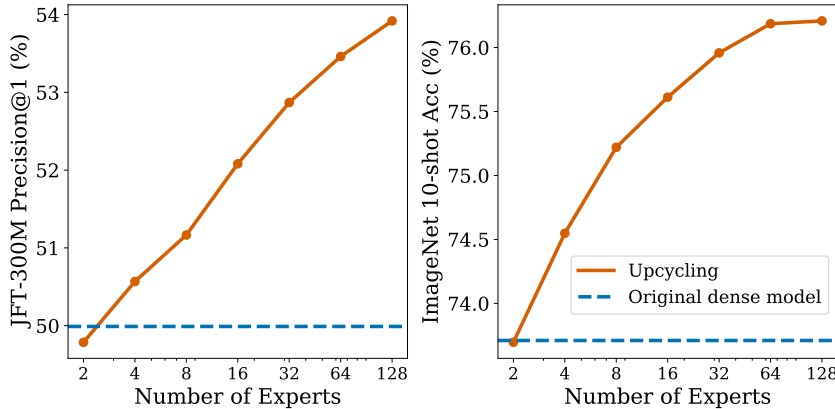

Figure 11: Final upstream and downstream performance for upcycled B/16 vision models with different number of experts per MoE layer. The number of MoE layers is fixed at 6. The upcycled model is trained for an additional 7 epochs (from 14 to 21) relative to the original dense model. The dashed horizontal lines show the performance of the dense model when trained for an additional 7 epochs.

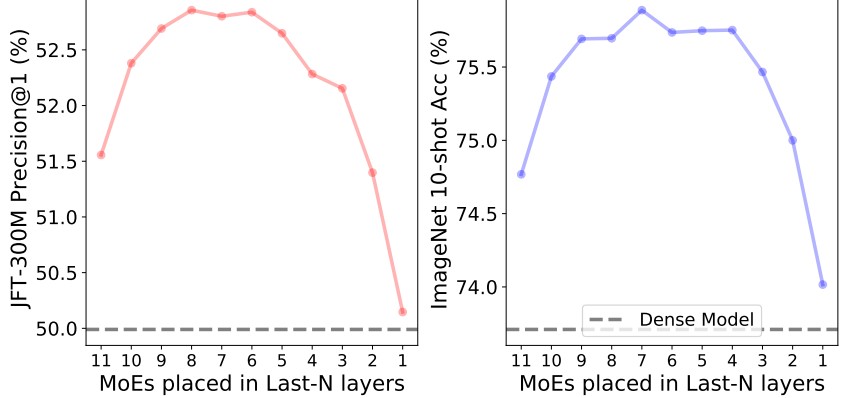

Figure 12: Performance as a function of the number of MoE layers for upcycled B/16 models ($C = 1$) trained for 7 additional epochs on JFT, starting from a dense checkpoint originally trained for 14 epochs. MoE layers are consecutively placed starting from the last block. We train models ranging from 1 MoE layer (Last-1) to 11 MoE layers (Last-11) – i.e. all but the very first. The dashed horizontal lines show the performance of the dense model when trained for an additional 7 epochs.

## B.5 EXPERT INITIALIZATION

The standard upcycling recipe copies and replicates the dense MLP to each expert. As the router directs different tokens to each expert, the experts will start to diverge from one another, and their initial MLP weights. Figure 13 explores whether loading the MLPs is indeed a good idea, or whether the model would be better off learning the experts from scratch (random initialization). We train for 7 extra epochs (dense was trained for 14 epochs, and we keep training up to a total of 21). Note that the computational cost of both approaches is identical.

It takes a long time for the model with randomly initialized experts to recover and catch up with the algorithm that upcycles the expert weights from the dense MLP layers, regardless of the number of experts. We also tried an intermediate approach (not shown), where we only upcycle a subset of experts and initialize the rest of scratch, but that also underperformed upcycling all of the experts.

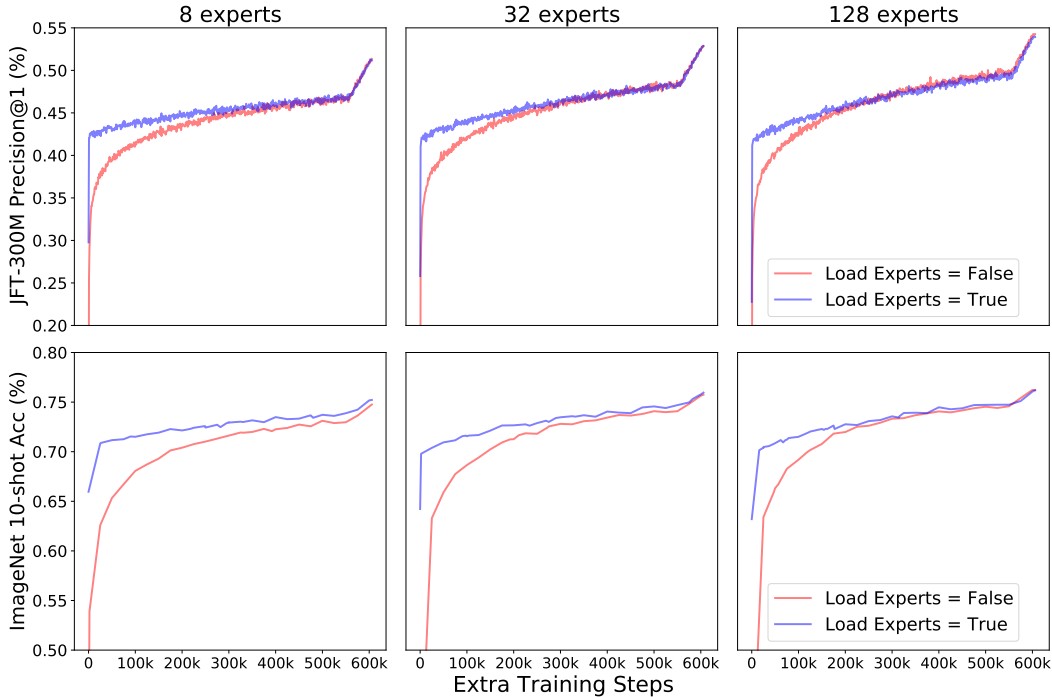

Figure 13: Performance comparison between upcycling experts ("Load Experts = True") and randomly initializing the experts ("Load Experts = False"). We include upstream (top row) and downstream (bottom row) performance metrics, and also ablate the number of experts per MoE layer (over the columns).

### B.6 RESUMING THE OPTIMIZER STATE

When upcycling a model, we can resume the optimizer state from the original dense checkpoint together with the model parameters. Figure 14 shows that reusing the optimizer state gives a performance boost for vision models, independent of the number of experts.[6] We did not, however, see any improvement from reusing the dense model optimizer state in our language experiments, so we only reuse the optimizer state for vision models.

### B.7 COMBINE WEIGHT NORMALIZATION AFTER ROUTING

A simple trick that we found useful for the upcycling of vision models was to normalize the combine weights after routing. The weights of each token are normalized so that the sum is 1. This follows that intuition that each token was previously only processed by a single "expert" MLP in the dense model. In the event that a token is not routed at all, the combine weights remain 0.

We illustrate this normalization trick with two simple examples.

**Several experts selected**. Suppose a token $x$ is selected by three different experts $e_1, e_2$ and $e_3$ with routing weights $w_1 = 0.3, w_2 = 0.2$, and $w_3 = 0.1$ respectively (adding up to 0.6).

The normalized weights are:

$$\bar{w}_1 = \frac{0.3}{0.6} = 0.5, \qquad \bar{w}_2 = \frac{0.2}{0.6} = 0.3333..., \qquad \bar{w}_3 = \frac{0.1}{0.6} = 0.1666...$$

The final output $x'$ is:

$$x' = \bar{w}_1 \cdot e_1(x) + \bar{w}_2 \cdot e_2(x) + \bar{w}_3 \cdot e_3(x).$$

---

[6]For some parameter, such as the router weights, we do not have any original optimizer state that we can reuse.

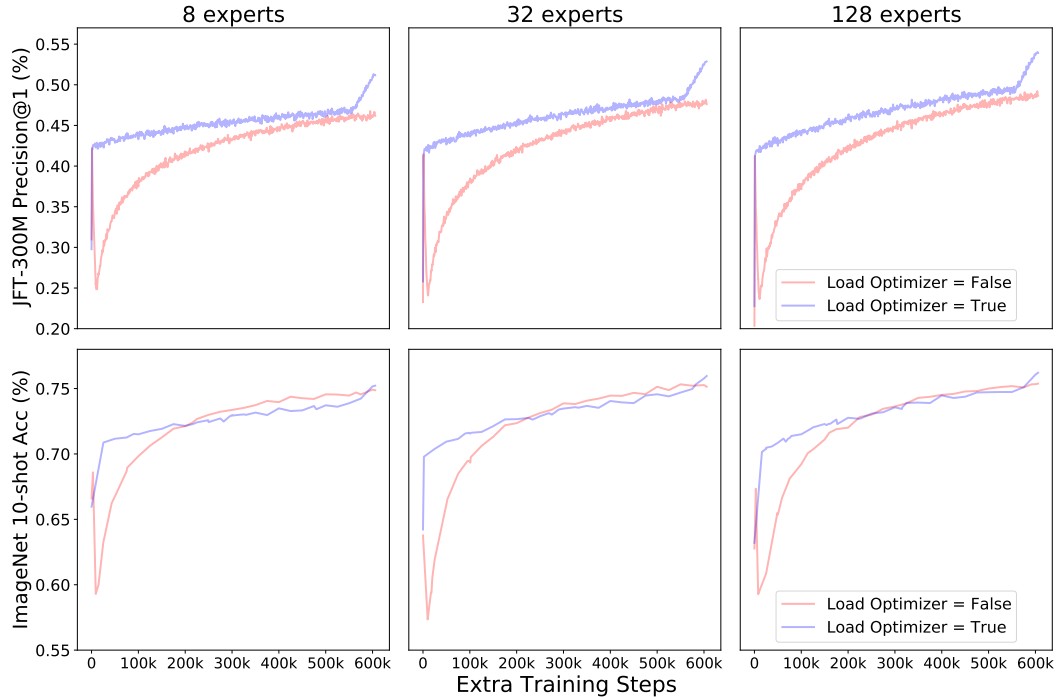

Figure 14: Performance comparison between reusing ("Load Optimizer = True") and not reusing ("Load Optimizer = False") the optimizer state. We include upstream (top row) and downstream (bottom row) performance metrics, and also ablate the number of experts per MoE layer (over the columns).

Table 3: Training from scratch on V-MoE-B/32 vision models with Expert Choice routing. Comparison with and without weight renormalization after routing.

| Capacity | Renormalization | Val Prec@1 | ImageNet 10shot |
|---|---|---|---|
| $C = 1$ | No | 48.71 | 69.68 |
| $C = 1$ | Yes | 48.23 | 70.19 |
| $C = 2$ | No | 50.02 | 71.26 |
| $C = 2$ | Yes | 49.75 | 71.55 |

**Only one expert selected**. In this case, regardless of the selected weight $w_1$, the output routing weight will be $\bar{w}_1 = 1.0$ after normalizing it:

$$x' = \bar{w}_1 \cdot e_1(x) = 1.0 \cdot e_1(x).$$

While this approach can be in principle a bit problematic (those tokens only selected by one expert have vanishing routing gradients), Table 3 shows that, even if we are training vision models from scratch, applying weight normalization does not hurt performance (while it indeed helps for upcycling).

However, router weight normalization was not helpful for language models. Upstream accuracy after 1M steps was comparable: 70.8% (no normalization) vs 70.7% (normalization), but downstream average scores on SuperGLUE lagged: 79.3% (no normalization) vs 78.8% (normalization). A similar quality degradation were observed in MoE language models trained from scratch. One hypothesis for this different behavior is that the vision models use Expert Choice routing everywhere, but the language models use Expert Choice in the encoder and Top-K routing in the decoder.

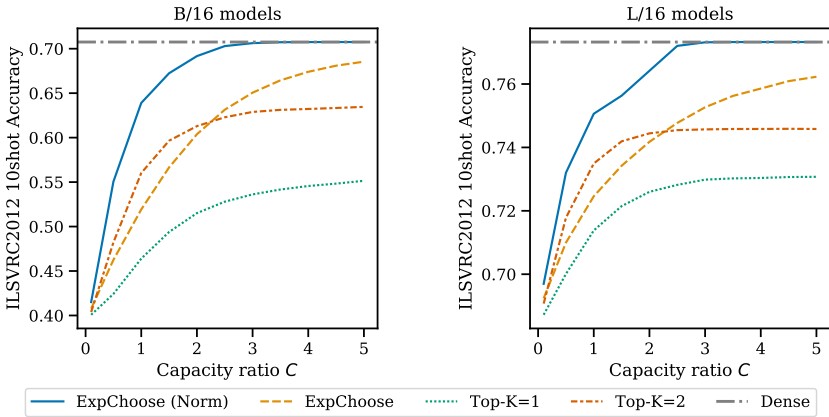

Figure 15: The effect of capacity size ratio on the initial performance of B/16 (left) and L/16 (right) models after upcycling (i.e. at the very first new step); when routing weights are normalized (Section B.7), and capacity is large, the upcycled model retains the dense model's function.

## B.8 CLOSER LOOK AT DESIGN CHOICES IMMEDIATELY FOLLOWING UPCYCLING

Once a model is upcycled, there is a drop in performance due to the sudden deviation from the dense model's learned function, even with MoE experts reusing the dense MLP weights. Routing design decisions can make a significant difference in ameliorating this drop.

Section B.2 ablates the long-term upcycled model performance as a function of the capacity factor $C$. Figure 15 shows the immediate effect of modifying $C$ on the upcycled model. Increasing $C$ reduces the likelihood that tokens are dropped; when routing weights are normalized to sum to 1 (see Section B.7), the upcycled model is exactly equivalent to the dense model for those tokens selected by at least one expert. Note the set of tokens not selected by any expert decreases in size as we increase $C$.

Note that although different routing mechanisms significantly impact the starting point for upcycling, the subsequent training can smooth over many differences. For example, at the start Top-K routing is clearly worse than Expert Choice with weight normalization, but Table 2 shows that a model upcycled with Top-K routing eventually catches up.

Figure 16 shows the effect of the group size parameter. Routing, and in particular the top-k operations, are performed in groups. Using smaller group sizes will speed up routing, but will lead to higher variance in expert assignment across groups. For smaller groups, we may expect more tokens to be dropped (not selected by any expert).

How many MoE layers we upcycle, and where we place those layers, also affects the initial performance drop. Figure 17 shows the effect of this in the initial drop for Expert Choice routing, using normalized combined weights (Section B.7) Upcycling the bottom layers causes a larger initial performance drop. Upcycling the last layers consecutive layers or interleaving them –as in every other layer– yields the smallest initial performance drop.

Finally, we analyze how the number of experts per MoE layer affects the initial upcycling. Figure 18 suggests that routing to more experts leads to a heavier drop. Figure 12 shows that upcycled models can recover from this eventually though, and sometimes even achieve higher performance.

## B.9 THINGS THAT DID NOT WORK

We list several unsuccessful attempts, beyond the preceding ablations, to improve the performance of the upcycling. Adding (truncated) Gaussian noise to router weights, expert weights or both, in an effort to diversify the router or expert weights, did not help. Modifying the learning rate of experts, routers or both, to account for the fact that the addition of sparsity may demand different learning rate from the rest of the model, generaly hurt performance; increasing the learning rates sometimes rendered the models more unstable. We attempted to add a fixed temperature term to the softmax of

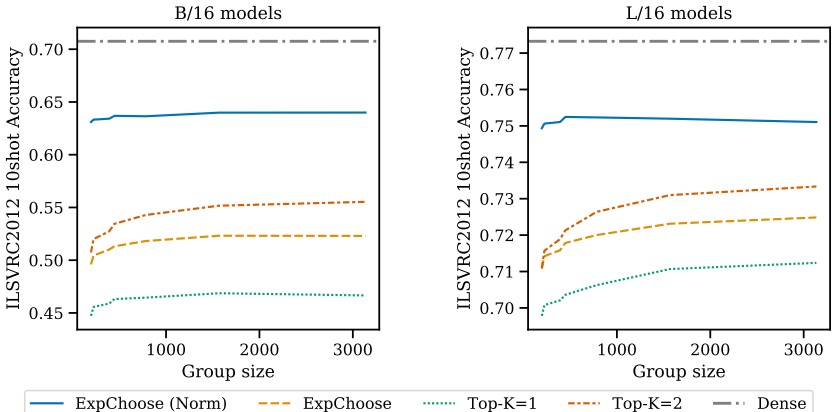

Figure 16: Increasing group size does not significantly affect performance using Expert Choice routing, but improves initial performance for Top-K routing.

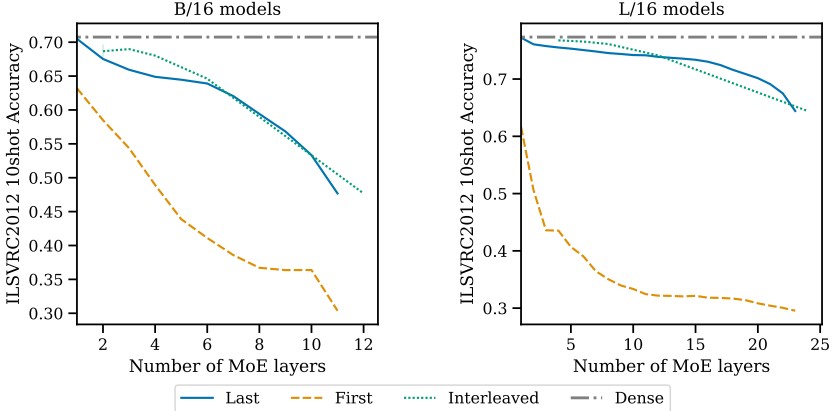

Figure 17: Effect of position and number of MoE layers on the initial performance after upcycling (i.e. at the very first new step). Note that L/16 models have more MLP layers..

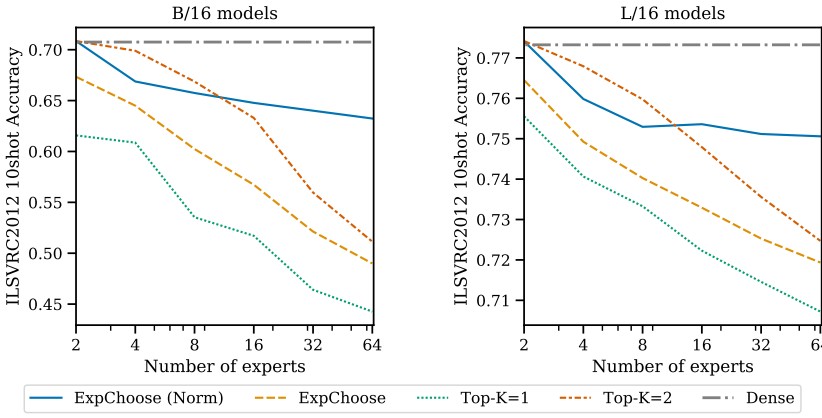

Figure 18: Effect of the number of experts per MoE layer on the initial drop of ImageNet 10-shot performance (i.e. at step 1), for B/16 and L/16 upcycled models with 6 MoE layers and $C = 1$. The initial performance is lower when using more experts.

router to encourage more random or less random routing. We also varied the initialization scheme for the router, included the router z-loss (Zoph et al., 2022). Unfortunately, none of these attempts led to any significant performance improvement.

# C  SELECTED RESULTS

Table 4: Selection of results on vision tasks for different methods and architecture variants. We report the JFT-300M Validation Precision at 1 (%), the ILSVRC2012 10shot Accuracy (%), the ILSVRC2012 Accuracy after Finetuning (%), and the extra TPUv3-core-days and ExaFLOPs used, both in absolute and relative (%) terms with respect to the corresponding dense checkpoint (when these are 0, they correspond to the initial dense checkpoints).

| Method | Variant | JFT-300M | ILSVRC2012 10shot | ILSVRC2012 Finetune | Extra TPUv3-days | Relative Extra TPUv3-days | Extra ExaFLOPs | Relative Extra ExaFLOPs |
|---|---|---|---|---|---|---|---|---|
| Dense | B/32 | 37.78 | 61.92 | 78.91 | 0.00 | 0.00 | 0.00 | 0.00 |
| Dense | L/32 | 44.69 | 70.97 | 83.08 | 0.00 | 0.00 | 0.00 | 0.00 |
| Dense | B/16 | 46.67 | 72.19 | 83.77 | 0.00 | 0.00 | 0.00 | 0.00 |
| Dense | L/16 | 53.54 | 78.63 | 86.50 | 0.00 | 0.00 | 0.00 | 0.00 |
| MoE | B/32 | 20.24 | 39.82 | 66.95 | 1.91 | 14.60 | 4.02 | 14.64 |
| Upcycling | B/32 | 38.73 | 63.01 | 79.58 | 2.53 | 19.31 | 5.32 | 19.36 |
| MoE | L/32 | 23.76 | 44.63 | 69.11 | 5.70 | 14.11 | 13.79 | 14.36 |
| Upcycling | L/32 | 45.33 | 71.28 | 83.35 | 5.78 | 14.31 | 13.15 | 13.69 |
| MoE | B/32 | 38.77 | 63.24 | 78.80 | 9.57 | 73.01 | 20.12 | 73.19 |
| Upcycling | B/16 | 47.81 | 73.19 | 84.13 | 12.36 | 13.54 | 28.31 | 12.68 |
| Upcycling | B/32 | 46.34 | 69.98 | 82.18 | 21.60 | 164.88 | 45.43 | 165.30 |
| Upcycling | L/16 | 53.89 | 78.78 | 86.56 | 33.99 | 10.87 | 81.36 | 10.40 |
| Upcycling | L/32 | 51.19 | 74.43 | 84.66 | 49.95 | 123.58 | 113.63 | 118.29 |
| Dense | B/16 | 48.12 | 73.28 | 84.23 | 53.19 | 58.26 | 130.04 | 58.26 |
| MoE | B/16 | 50.24 | 74.96 | 84.57 | 73.58 | 80.59 | 168.49 | 75.49 |
| Upcycling | B/16 | 52.59 | 76.48 | 85.67 | 85.94 | 94.13 | 196.80 | 88.17 |
| Upcycling | B/16 | 54.42 | 77.39 | 86.03 | 159.52 | 174.73 | 365.28 | 163.66 |
| Dense | L/16 | 54.91 | 79.17 | 86.68 | 182.16 | 58.26 | 455.85 | 58.26 |
| Upcycling | L/16 | 56.71 | 79.70 | 87.10 | 236.27 | 75.57 | 565.58 | 72.29 |
| Dense | L/16 | 55.65 | 79.46 | 86.48 | 338.49 | 108.26 | 847.05 | 108.26 |
| Upcycling | L/16 | 57.84 | 80.01 | 87.17 | 438.55 | 140.27 | 1049.79 | 134.18 |
| Dense | L/16 | 56.19 | 79.63 | 86.89 | 494.81 | 158.26 | 1238.24 | 158.26 |

Table 5: Selection of results on text tasks for different methods and architecture variants. We report the C4 Validation Token Accuracy, the individual metrics on each SuperGLUE task after finetuning, as well as the overall SuperGLUE score, and the extra TPUv4-core-days and ExaFLOPs used, both in absolute and relative (%) terms with respect to the corresponding dense checkpoint (when these are 0, they correspond to the initial dense checkpoints).

| Method | Variant | C4 | BoolQ | CB | COPA | MultiRC | ReCoRD | RTE | WiC | WSC | SuperGLUE Score | Extra TPUv4-days | Relative Extra TPUv4-days | Extra ExaFLOPs | Relative Extra ExaFLOPs |
|---|---|---|---|---|---|---|---|---|---|---|---|---|---|---|---|
| Dense | Base | 67.97 | 82.48 | 94.64 / 93.69 | 70.00 | 38.30 / 76.52 | 77.26 / 78.24 | 82.31 | 69.59 | 83.65 | 77.17 | 0.00 | 0.00 | 0.00 | 0.00 |
| Dense | Large | 71.80 | 87.95 | 96.43 / 95.03 | 87.00 | 54.35 / 84.35 | 84.84 / 85.78 | 92.06 | 75.55 | 87.50 | 85.06 | 0.00 | 0.00 | 0.00 | 0.00 |
| Dense | XL | 74.15 | — | — | — | — | — | — | — | — | — | 0.00 | 0.00 | 0.00 | 0.00 |
| Dense | Base | 68.20 | 81.77 | 92.86 / 91.79 | 70.00 | 37.88 / 76.04 | 77.25 / 78.15 | 82.67 | 67.55 | 81.73 | 76.34 | 13.33 | 40.00 | 55.15 | 40.00 |
| Dense | Base | 68.33 | 83.00 | 94.64 / 93.61 | 72.00 | 38.93 / 76.82 | 77.46 / 78.49 | 84.84 | 67.71 | 78.85 | 77.05 | 19.99 | 60.00 | 82.73 | 60.00 |
| Upcycling | Base | 69.78 | 82.91 | 98.21 / 98.68 | 74.00 | 38.09 / 77.20 | 80.90 / 81.95 | 82.67 | 69.28 | 87.50 | 79.23 | 20.31 | 60.98 | 82.14 | 59.57 |
| MoE | Base | 69.52 | 79.66 | 91.07 / 89.08 | 63.00 | 30.12 / 73.44 | 76.36 / 77.48 | 79.78 | 69.75 | 84.62 | 74.45 | 20.33 | 61.01 | 82.14 | 59.57 |
| Upcycling | Base | 70.22 | 84.04 | 100.00 / 100.00 | 76.00 | 40.08 / 78.20 | 82.06 / 83.05 | 83.39 | 68.03 | 84.62 | 79.72 | 30.47 | 91.47 | 123.21 | 89.36 |
| MoE | Base | 70.12 | 80.34 | 96.43 / 97.38 | 72.00 | 30.33 / 73.87 | 77.43 / 78.66 | 80.51 | 70.06 | 84.62 | 76.82 | 30.49 | 91.52 | 123.21 | 89.36 |
| Dense | Base | 68.53 | 82.63 | 96.43 / 93.21 | 70.00 | 38.30 / 77.27 | 77.55 / 78.46 | 83.03 | 69.91 | 82.69 | 77.36 | 33.32 | 100.00 | 137.88 | 100.00 |
| Upcycling | Base | 70.83 | 84.31 | 98.21 / 98.68 | 79.00 | 39.77 / 77.84 | 83.02 / 84.15 | 83.39 | 69.59 | 81.73 | 79.86 | 50.79 | 152.45 | 205.35 | 148.94 |
| Dense | Large | 72.03 | 88.81 | 96.43 / 94.30 | 89.00 | 56.87 / 85.49 | 87.65 / 88.44 | 90.97 | 73.20 | 90.38 | 85.87 | 123.54 | 40.00 | 637.72 | 40.00 |
| Dense | Large | 72.12 | 88.81 | 96.43 / 94.30 | 88.00 | 56.56 / 85.54 | 87.78 / 88.54 | 90.25 | 73.35 | 90.38 | 85.67 | 185.42 | 60.03 | 956.58 | 60.00 |
| Upcycling | Large | 73.34 | 88.62 | 96.43 / 95.04 | 90.00 | 55.93 / 85.22 | 87.65 / 88.80 | 90.61 | 72.26 | 92.31 | 86.04 | 197.38 | 63.91 | 1076.51 | 67.52 |
| Upcycling | Large | 73.68 | 88.90 | 100.00 / 100.00 | 90.00 | 55.40 / 85.04 | 88.58 / 89.62 | 92.78 | 72.57 | 90.38 | 86.74 | 287.48 | 93.08 | 1614.73 | 101.28 |
| Dense | Large | 72.26 | 88.38 | 96.43 / 94.30 | 87.00 | 57.08 / 85.43 | 87.83 / 88.57 | 91.34 | 72.57 | 87.50 | 85.20 | 309.04 | 100.06 | 1594.30 | 100.00 |
| Dense | XL | 74.34 | — | — | — | — | — | — | — | — | — | 508.73 | 40.00 | 2369.07 | 40.00 |
| Dense | XL | 74.46 | — | — | — | — | — | — | — | — | — | 1050.73 | 82.62 | 3401.70 | 57.44 |
| Upcycling | XL | 75.03 | — | — | — | — | — | — | — | — | — | 1405.57 | 110.52 | 4985.16 | 84.17 |