# OpenReview forum: "Sparse Upcycling: Training Mixture-of-Experts from Dense Checkpoints"
_ICLR.cc/2023/Conference — ICLR 2023 poster_

### Official Review · Reviewer_vrXq · 2022-10-24

**Confidence:** 5
**Clarity, Quality, Novelty And Reproducibility:** see above
**Correctness:** 3
**Technical Novelty And Significance:** 3
**Empirical Novelty And Significance:** 3
**Recommendation:** 5

**Strength And Weaknesses:**

The paper proposes a training strategy to make MoE training faster. The proposed sparse upcycling is very reasonable. The experiments also show the superiority of this training strategy.

I have the following concern and doubts:

1. The proposed sparse upcycling only increases training speed, but not performance. As shown in Figure.4 right, upcycling and moe from scratch would finally converge.

2. The proposed method seems to take as long as the moe from scratch to finally converge. In Fig.4 and Fig.5, the training curve is not finally converged so it occurs to me that sparse upcycling are just faster in the early stage but could take as long as the moe from scratch to converge.

3. The training speed still seems to be at the same level as training a normal moe. The improvement in time is not a game changer.

4. In the ideal case, MoE only increases the params size but not computation cost or computation speed. (which depends on topk) So the dense model training time should be similar to the sparse model training time even though the sparse model is much larger. I am not sure if that really matters to increase the training speed. A demonstration of the relation between params size, computation cost, and the training speed using normal training or the proposed upscale training is missing from the paper. In another word, under what circumstances (params size, FLOPS) do we care about the training speed, and how much this method can improve?

Above all, the method is simple and solid. My major concern is that the speed improvement does not seems to be significant and crucial in a real application and the paper does not demonstrate why this is useful. I am also not sure if upcycling is on a different speed level of a normal MoE with decent optimization.



----------Afrter Rebuttal---------------

I have read the response from the authors. I still have doubts about the speed improvement: not likely to be significant and mostly comes from the slow MoE implementation. I would keep my rating.


**Summary Of The Paper:**

The paper proposes sparse upcycling -- copy the model into a sparse model for faster and more efficient training. They conduct experiments on JFT-300M and English C4 dataset. The empirical results show its superiority.

**Summary Of The Review:**

see above

---

> ### Author Response · Authors · 2022-11-18
> **Response to Reviewer vrXq**
>
> Thank you for your detailed review. We plan to update the paper (in particular the Introduction) to reflect the below discussion prompted by the reviewer.
>
> We agree with the reviewer that sparse upcycling primarily affects the training speed. The performance is certainly improved relative to continuing to train the dense model (see Figure 2). Comparing the MoE-from-scratch and sparsely upcycled model (Figure 4), we indeed see that the from-scratch model eventually catches up to the upcycled model and potentially – for language at least – converges to a better accuracy. Given an unlimited training budget, this is something reasonable that we expected.
>
> Our disagreement with the reviewer rests on the importance of training speed. We argue that, in practice, training speed is critical in many settings. For example, when (1) researchers iterate through models and hyperparameter settings; (2) engineers refresh production models on new data; and (3) we, as a community, seek to lower the energy costs of models (which is proportional to the training time).
>
> In the new version, we have also further emphasized the regimes where sparse upcycling is helpful (20-100% of initial compute budget) vs regimes where it is not (unlimited compute / training until convergence); see Section 4.2.1. Indeed, sparse upcycling is primarily useful when the compute budget is constrained and training the MoE model to convergence is just too expensive; e.g. when the model and/or dataset is too large. In such cases, given some modest additional computational budget, what is the most efficient use of your budget?
>
> Responding further to specific comments:
> 1. As alluded to above, real applications are often constrained by available accelerator training time/budgets and production model “freshness”.
> 2. We agree that given a large computation budget (> 100% of initial compute for Figure 5), the MoE-from-scratch model eventually catches the upcycled model, at least for the language task (we strongly suspect the same would happen for vision). For such large computation regimes, we suggest training MoE models from scratch. For constrained or limited compute budgets (< 100% of initial compute), we propose sparse upcycling. We have clarified this framing in our comparison of MoE-from-scratch with sparse upcycling in Section 4.2.1.
> 3. Note that in Figure 4 (upcycling vs MoE-from-scratch), the upcycled model shows a significant quality gap over MoE-from-scratch for constrained compute budgets (e.g. <= 50 % of the initial pre-training compute budget).
> 4. See discussion above. In practice, parameter size is typically only a major concern in memory constrained settings. FLOPS provides an imperfect proxy for speed [1]. As argued above, “computation cost” (training speed x training time) is a major consideration in many real applications. As an aside, note that because we use an expert capacity factor = 2 (argued to be optimal in Section 4.2.2 and Appendix B.2), the dense and MoE models have different run/training speeds.
>
> [1] Mostafa Dehghani, Anurag Arnab, Lucas Beyer, Ashish Vaswani, Yi Tay, The Efficiency Misnomer, https://arxiv.org/abs/2110.12894 (2022)

---

### Official Review · Reviewer_puDi · 2022-10-24

**Confidence:** 4
**Correctness:** 3
**Technical Novelty And Significance:** 3
**Empirical Novelty And Significance:** 4
**Recommendation:** 8

**Clarity, Quality, Novelty And Reproducibility:**

The proposed idea is not entirely new and it has been a practice to initialize MOE models with dense checkpoints. However, the insights from this work are of high quality which provides some important design choices when there is limited computation budget and we want to get better performance. The ablation studies have clearly demonstrated the design choices.

In terms of reproducibility, for the sparse models in this work, will depend on the availability of the pretraining datasets like JFT. However, the learnings from this work will be widely applicable and the thorough discussion of the design choices will help reproducing this work on other datasets. I would encourage the authors to release the code/model/data as far as possible to make this study reproducible.

**Strength And Weaknesses:**

### Strengths
- Sparse upcycling clearly shows improved performance for both vision and language pretraining, with significant improvements when some non-trivial computation budget is available for further training.
- importance of this paradigm is in general great to reuse/recycle old models
- Performance improvement is fairly consistent irrespective of how long the dense pretraining was done, which is promising.
- Authors discuss various design choices for upcycling clearly. Ablation experiments are thorough and clearly justifies different design choices.

### Weaknesses
- Initializing MOE models from dense checkpoints is not novel and is somewhat a known practice.
- Figure 3 is missing the finetuning performance for XL models for language tasks.
- The performance on downstream language tasks has a lot of variance which increases with larger models and somewhat seems that upcycling does not help in SuperGLUE tasks to a large extent. Is this dependent on downstream tasks? Is this behavior same on other downstream language tasks as well?


**Summary Of The Paper:**

The paper presents an interesting and useful idea for reusing pretrained dense checkpoints to initialize larger sparse models for better performance on different vision and language tasks. This method helps to improve performance with some additional computational budget but does not require training large sparse models from scratch. The paper conducts analysis on T5 and ViT models with various sizes and discusses different design decisions for upcycling.

**Summary Of The Review:**

Overall, this is an interesting work which shows that we can get improved performance by initializing sparse models with dense pretrained checkpoints when there is additional computational budget available. Initialization of sparse models from dense checkpoints is not new to the community but this work shows some good design choices which are important and will be useful. Hence I recommend this rating, and look forward to the author discussion phase.

---

> ### Author Response · Authors · 2022-11-18
> **Response to Reviewer puDi**
>
> Thank you very much for your review.
>
> We agree that the core idea of upcycling a dense model into a sparse one is relatively simple. However, as far as we are aware, the majority of works that initialize MoE models from dense checkpoints do so through more complicated or multistage end-to-end training schemes; see, for example, [1] (also cited in our paper). Another relevant work is [2], which is discussed in some detail in a separate comment thread.
>
> The SuperGLUE accuracies of both the dense and sparse models are indeed noisy. For the vision tasks, we averaged results over multiple runs to lower the variance in accuracy. We hadn’t yet done this for language because both the task dataset and the models are larger (SuperGLUE is a suite of tasks), but we are currently running more SuperGLUE trials so that we can report average scores. These should be complete for a camera ready version. That said, we believe that the trend between the sparse and dense models is clear even with this noise.
>
> For similar computational reasons, we have not yet fine-tuned the T5 XL model across multiple data points, although we are trying to get our hands on more resources!
>
> We plan to release both the language and vision code.
>
> [1] Xiaonan Nie, Xupeng Miao, Shijie Cao, Lingxiao Ma, Qibin Liu, Jilong Xue, Youshan Miao, Yi Liu, Zhi Yang, Bin Cui, EvoMoE: An Evolutional Mixture-of-Experts Training Framework via Dense-To-Sparse Gate, https://arxiv.org/abs/2112.14397 (2021)
>
> [2] Lemeng Wu, Mengchen Liu, Yinpeng Chen, Dongdong Chen, Xiyang Dai, Lu Yuan, Residual Mixture of Experts, https://arxiv.org/abs/2204.09636 (2022)

---

### Official Review · Reviewer_36d4 · 2022-10-25

**Confidence:** 4
**Clarity, Quality, Novelty And Reproducibility:** The paper is clear and the idea is no…
**Correctness:** 4
**Technical Novelty And Significance:** 3
**Empirical Novelty And Significance:** 3
**Recommendation:** 8

**Strength And Weaknesses:**

The paper is generally clear and easy to follow. The problem of using already trained dense checkpoints for initializing sparse models is important, especially given the promising results of sparse MoE models in different domains. I found the idea of replicating the MLP layer(s) and randomly initializing the routing mechanism simple yet powerful. I have the following questions and comments:
1. How do the distribution of token to expert assignment look like? Are we seeing few experts which dominate the probability distributions?
2. How many experts are involved in processing each token (on average)?
3. How do the self-similarities and intersimilarities of inputs between experts look like?
4. The related work, is missing some works on converting a dense model to its corresponding MoE version. See for instance:
    1. MoEfication: Transformer Feed-forward Layers are Mixtures of Experts
    2. MoEBERT: from BERT to Mixture-of-Experts via Importance-Guided Adaptation
    3. DEMIX Layers: Disentangling Domains for Modular Language Modeling (not directly related but uses a similar idea of replicating the MLP units).
5. It would be nice to add a comparison to one of the aforementioned approaches to show the advantages of the proposed method.


**Summary Of The Paper:**

The paper proposes an approach for initializing sparse Mixture of Expert (MoE) models from dense checkpoints. The approach called upcycling reuses an already trained dense checkpoint by copying all the parameters from the original checkpoint (except for the MoE router parameters). The experts in this new model are replicas of the original MLP layer(s) in the dense model. The experiments demonstrate that the upscaled models outperform the sparse models trained from scratch when using the same computation budget as the initial dense pretraining.

**Summary Of The Review:**

Overall, I like the idea and the fact that the paper covers a relatively comprehensive set of experiments for both vision and language domains.

---

> ### Author Response · Authors · 2022-11-18
> **Response to Reviewer 36d4**
>
> Thank you for your review. We reply to your specific questions below.
>
> 1. > “How do the distribution of token to expert assignment look like?”
>
> We use Expert Choice routing [1], where each expert selects a subset of the tokens. The distribution of how many experts process each token varies per MoE layer. As an example, for an upcycled Vision B/16 with last-6 MoE layers (11th-16th layers), we see the following at the 11th layer.
>
> C = 1 (each expert processes 1/E fraction of the tokens):
> % of tokens selected by at least 1 expert: between 81% and 85%.
> % of tokens selected by at least 2 experts: between 13% and 18%.
> % of tokens selected by at least 4 experts: less than 0.1%.
> % of tokens selected by at least 8 experts: 0%.
>
> C = 2 (each expert processes 2/E fraction of the tokens):
> % of tokens selected by at least 1 expert: >98%.
> % of tokens selected by at least 2 experts: between 68% and 71%.
> % of tokens selected by at least 4 experts: between 4% and 5%.
> % of tokens selected by at least 8 experts: 0%.
>
> Most tokens are processed roughly C times and a few tokens are processed either by a few experts (usually not large deviations) or by none at all (~15-20% in the case of C=1).
>
> 2. > “Are we seeing few experts which dominate the probability distributions?”
>
> We did not see this. As Expert Choice routing balances all experts by design, and experts are already “well-developed” (as they come from the pre-trained checkpoint), we suspect that the difference in “importance” across experts (as, measured, for instance by the average routing weight per expert before applying top-k) is not very large.
>
> 3. > “How many experts are involved in processing each token (on average)?”
>
> The numbers above should give a sense of the expert-per-token distribution. The average, by definition, is C, as each expert always processes C * T / E tokens, where T is the total number of tokens.
>
> 4. > “How do the self-similarities and intersimilarities of inputs between experts look like?”
>
> We haven’t explored this in detail, while it is a very interesting aspect and we have discussed it a few times. However, we suspect that experts from upcycled models will tend to be more similar to each other than those trained from scratch, as it takes time to “move away” from each other. We considered, but did not explore, adding an auxiliary loss that explicitly forces the experts to be dissimilar – we generally believe it is better not to add inductive biases unless absolutely necessary (e.g. balancing loss when TokensChoose routing is used in the decoder).
>
> 5. Thank you for the relevant references! We have added all of these to the paper.
>
> 6. > Comparison with related work
>
> One of the closest related works is the warm starting / parameter re-use techniques (“dense upcycling”) explored in Gopher [2], where parameters from a smaller model are used to initialize a larger model by tiling weights depthwise (layer) or widthwise (model dimension). We have added a comparison of sparse upcycling with the parameter re-use from Gopher [2] to Section 4.2.1. Summarizing our findings: though Gopher warm-starting improves over the original dense checkpoint, it clearly underperforms the sparsely upcycled model by a large margin.
>
> [1] Zhou et al, Mixture-of-experts with expert choice routing, https://arXiv:/2202.09368, (2022)
>
> [2] Rae et al., Scaling language models: Methods, analysis & insights from training gopher, https://arXiv:2112.11446, (2021)

---

### Official Review · Reviewer_zFGk · 2022-10-25

**Confidence:** 4
**Correctness:** 3
**Technical Novelty And Significance:** 3
**Empirical Novelty And Significance:** 3
**Recommendation:** 6

**Clarity, Quality, Novelty And Reproducibility:**

- The paper is clearly presented, and extensive experiments verified the efficacy;
- the idea of using dense for upcycling is novel and effective;
- the author promises to release the upcycle code upon acceptance.

Typo - Appendix B.1 "on a per step basis, provided we also use Batch Priority Routing (BPR)" -> "on a per step basis when using Batch Priority Routing (BPR)"

**Strength And Weaknesses:**

Summary Of Strengths
- the paper is clearly written and presented;
- the efficacy of the sparse upcycling is verified both across domain (vision & language), transfer (downstream) and scales (up to 3B);
- extensive ablations are provided (router, #dense pretrain, #MoE layers, #experts, MoE locations, and initialization) to illustrate the design choices, which may pave the road for future researchers and practitioners in the efficient computing area.

Summary Of Weaknesses
- Fairer comparison: it is clearly presented in figure 2 that a sparsely upcycled model outperforms a continual dense counterpart on vision, and language pretraining. However, the sparsely upcycled model also has more capacity, I am wondering if will it be a fairer comparison for:

  (i) upcylce a dense model to a large model (though it will introduce more inference cost) as in [1,2,3,4], especially [1,2] show that staged warm starting may also help the efficiency of training;

  (ii) continue to train a MoE model as shown in Figure 4, and why the Appendix Table 2 seems to disagree what presented in Figure 4?
  7+7 extra epochs of MoE models seem to outperform a sparsely upcylced model with 7+ extra epochs.

- Why continual training of a dense model turns to hurt the downstream performance on SuperGLUE;
- How sensitive are the design choices like in the following will affect (1) the initial transition; (2) the ending performance?

    (i) the learning rate schedule: resuming/restarting [1], (also in Appendix A.1.1 suggest using a learning rate schedule where the dense checkpoint left off, does that suggest you will initially plan for more epochs, what if the learning rate has already decay to 0)

    (ii) MoE losses: [5] uses half loading balance + half importance and strong regularization like noise jitter and larger capacity ratio to make the V-MoE stable. Will all the techniques (except for the BPR) also be used here and will all these hyperparameters also be sensitive.

- the intuition in Figure 4 that why will a upcycled model will outperform continue-training MoE

   In Figure 3 and Appendix that the initial transition of the sparsely upcycled model underperforms a dense model, which I assume will also underperform a MoE model. Then using the same schedule, will the sparsely upcycled model consistently underperform a MoE counterpart that trained from scratch?


- Is it possible to upcycle MoE models?

[1] Rae, Jack W., et al. "Scaling language models: Methods, analysis & insights from training gopher." arXiv preprint arXiv:2112.11446 (2021).
[2] Shen, Sheng, et al. "Staged Training for Transformer Language Models." ICML 2022.
[3] Gu, Xiaotao, et al. "On the transformer growth for progressive bert training." NAACL 2021.
[4] Gong, Linyuan, et al. "Efficient training of bert by progressively stacking." ICML 2019.
[5] Riquelme, Carlos, et al. "Scaling vision with sparse mixture of experts." Advances in Neural Information Processing Systems 34 (2021): 8583-8595.



**Summary Of The Paper:**

This paper presents a novel yet simple method to reuse pretrained dense Transformer checkpoints to initialize larger sparse
models.
The authors showcase that using this technique will result in a more performant MoE model versus continuing training the dense counterparts on vision and language domains up to 3B scale.
Extensive ablations are provided for the design choices.


**Summary Of The Review:**

it is no doubt that sparse upcycling tries to leverage the inference & scaling benefits from MoE and pre-trained weight from a smaller Dense model, and it achieves better results given the more introduced capacity;

The major concerns for me are the inherited instability of training MoE also appears/worse in upcycling and will that affect the initial transition and how will the initial transition affect the end performance;

More details for the hyperparameter sensitivity will also be valuable to add.

---

> ### Author Response · Authors · 2022-11-18
> **Response to Reviewer zFGk**
>
> Thank you for your detailed review.
>
> > “Fair comparison with larger dense model.”
>
> We have added a comparison with larger, dense language models warm-started (“densely upcycled”) from smaller dense models, following the methodology in [1]. These new results appear as Figure 5 in Section 4.2.1. Summarizing our findings: the densely upcycled model quickly sees gains over the original dense checkpoint, but clearly underperforms the sparsely upcycled model.
>
> > “Comparing MoE-from-scratch and sparsely upcycled models.”
>
> Figure 4 (“MoE-from-scratch vs sparsely upcycled”) shows that the MoE-from-scratch model eventually catches up to the upcycled model and potentially – for language at least – converges to a better accuracy. This suggests that, given a large amount of available computation time (>120% of the initial pre-training budget in this case), it may be better to train an MoE from scratch. We have clarified the scope of our work in the Introduction to reflect that sparse upcycling is primarily beneficial in constrained computation regimes (20-100% of initial budget).
>
> > “Table 2 vs Figure 4.”
>
> Table 2 and Figure 4 cannot be directly compared. Figure 4 shows the B/16 model trained over 14 epochs, while Table 2 shows the L/32 model trained over 7 and 14 epochs. More importantly, we assume that the 7 epochs invested in pre-training the dense checkpoint is a sunk cost; i.e. the dense checkpoint already exists. This sunk cost cannot be used by the MoE-from-scratch model. What we can say from Table 2 is that the MoE-from-scratch model trained for 7+7 epochs (200% of initial pre-training budget sunk for dense checkpoint), outperforms the sparsely upcycled model trained for 7 epochs (100% of initial budget).
>
> > “Variance in model performance on SuperGLUE.”
>
> The SuperGLUE accuracies of both the dense and sparse models are indeed noisy. For the vision tasks, we averaged results over multiple runs to lower the variance in accuracy. We hadn’t yet done this for language because both the task dataset and the models are larger (SuperGLUE is a suite of tasks), but we are currently running more trials so that we can report average scores. These should be complete for a camera ready version.
>
> Given that we don’t expect large downstream improvements from further pre-training dense models which are already nearly trained to saturation, we suspect that a few of the Large dense SuperGLUE scores are outliers. Nevertheless, the trend between the sparse and dense models is clear.
>
> > “Design choice questions”: "learning rates"
>
> All our existing dense checkpoints are pre-trained with inverse square root decays (as is now common), so the learning rate is never zero. We experimented with increasing and decreasing the learning rate, both generally and only for specific components, but we found this either rendered the model unstable or offered no performance benefits; see also discussion in Appendix B.9. If the learning rate is already zero in the initial checkpoint, it may be possible to upcycle with a small constant rate, but we have not experimented with that.
>
> > “MoE losses”
>
> The standard recipe for upcycling that we explore in the paper uses Expert Choice routing [3]. Accordingly, we do not use any auxiliary load balancing loss. In Table 2 in Appendix B, we show that typical Top-K routing (as in V-MoE [2]) also works well with BPR. In that case, we do apply the same auxiliary MoE losses as V-MoE (i.e. load and importance).
>
> > “The intuition in Figure 4 that why will a upcycled model will outperform continue-training MoE.”
>
> Perhaps we misunderstand this question, but the premise of this work is that the dense checkpoints are given (many are publicly available), but an MoE checkpoint is not provided. So our comparisons are between continue training a dense model, training an MoE from scratch or sparsely upcycling a model. In Figure 4, the upcycled model outperforms the MoE-from-scratch model, on a pre-training time basis, because the upcycled model has a “head start” as it is initialized from the dense checkpoint.
>
> > “Upcycling MoE models.”
>
> This is an interesting suggestion that we did not explore. It should be straightforward to replicate experts to grow a, say, 32 expert model into a 64 expert model. Possibly more useful would be growing a Base 32 expert model into a Large 32 expert model, although [1] showed that this is challenging even for dense models.
>
> > “Inherited instability of training MoE also appears/worse in upcycling.”
>
> We did not see any evidence for this in our experiments. The downstream performance of the language models on SuperGLUE is noisy, but that is not a stability issue.
>
> We have added ref: Shen et al., "Staged Training for Transformer Language Models".
>
> [1] Rae et al., Scaling language models: Methods, analysis & insights from training gopher, (2021)
>
> [2] Riquelme et al., Scaling vision with sparse mixture of experts, (2021)
>
> [3] Zhou et al, Mixture-of-experts with expert choice routing, (2022)

---

### Public Comment · ~Lemeng_Wu1 · 2022-11-08
**Exsiting work using dense checkpoint to train MoE**

Dear Authors and Reviewers

Thank you for presenting and reviewing this interesting work. I would like to bring your attention to the highly relevant work Residual Mixture of Experts [1], which has not been discussed or cited.

Residual Mixture of Experts proposes a training pipeline using the dense model checkpoint pretrained on ImageNet22k to expand the MoE model and further finetune the downstream tasks like segmentation or detection for better performance and similar training time compared with dense model. We copy the MLP module into k copies as the MoE initialization and propose our receipt for a better tuning for this expanded model on the downstream task.

It is interesting that this work extends this setup into extremely large-scale benchmarks and gets new findings in different aspects, however, since [1] is shown on arxiv for nearly half-year before the ICLR deadline, I do not think it is a co-current work that does not need to cite and discuss at all in this paper.

[1] Residual Mixture of Experts

Lemeng Wu, Mengchen Liu, Yinpeng Chen, Dongdong Chen, Xiyang Dai, Lu Yuan

https://arxiv.org/abs/2204.09636

---

> ### Author Response · Authors · 2022-11-18
> **Thank you for the very relevant reference**
>
> Thanks for the comment and the pointer, Lemeng.
>
> We were not aware of this paper at the time of the submission, and we acknowledge that it is a highly relevant piece of work. Accordingly, we have updated the paper with a proper citation and a detailed discussion of it.
>
> There are a number of significant differences between our work and [1], that we briefly discuss next. The main motivation differs; in our work we are mostly interested in resuming upstream training with limited budget whereas [1] is mostly focused on downstream sparsification. Model surgery is substantially more challenging in the former scenario due to the initial performance drop caused by the model change. Indeed, [1] also considers our setup (referred to as “Intermediate” in Figure 2). In order to avoid the initial performance drop after adding experts, [1] applies a forward dense mixture of experts (see equation (7) in [1] with StopGradient – all experts are applied to every example) that, unfortunately, is not computationally feasible at the large model scale that we consider. Instead, we handle this issue via fully-sparse Expert Choice routing (rather than Top-k used in [1]) and a renormalization trick (see Appendix B.7). Moreover, [1] addresses object detection and segmentation tasks, while our work is focused on language models and image classification. There are some additional technical details (how layers to sparsify are selected, whether or not to add noise, number of experts, etc.) where both approaches diverge too.
>
> [1] Lemeng Wu, Mengchen Liu, Yinpeng Chen, Dongdong Chen, Xiyang Dai, Lu Yuan, "Residual Mixture of Experts", https://arxiv.org/abs/2204.09636 (2022)

---

### Decision · Program_Chairs · 2023-01-20

**Decision:**

Accept: poster

**Justification For Why Not Higher Score:**

One major concern raised by the reviewer who gave a score of 5 is that the proposed sparse upcycling only increases training speed, but not the final performance if one has unlimited computation budget. This could limit the impact of the work, so the AC thinks that it can be  a good poster paper, but not to the level of spotlight.

**Justification For Why Not Lower Score:**

This paper is overall well written and presented. The idea is simple but effective, experiments are also extensive. So, the AC thinks that it is above the ICLR acceptance bar.

**Metareview: Summary, Strengths And Weaknesses:**

This paper presents a simple method to reuse pretrained dense transformer checkpoints to initialize larger sparse models.

After author rebuttal, this paper received scores of 5688. All the reviewers agree that this paper is clearly written and presented. The idea is simple but effective. Experiments and ablations are comprehensive. On the other hand, one major concern raised by the reviewer who gave a score of 5 is that the proposed sparse upcycling only increases training speed, but not performance. During rebuttal, the authors have tried to make this clear that sparse upcycling is primarily useful when the compute budget is constrained.

Overall, given the overall positive reviewer comments, the AC would like to recommend acceptance of the paper.

**Note From Pc:**

if the above contains the word "oral" or "spotlight" please see: "oral" presentation means -> notable-top-5% and "spotlight" means -> notable-top-25%. As stated in our emails, we are disassociating presentation type from AC recommendations